# The Pareto Frontier of model selection
# for general Contextual Bandits

**Teodor Marinov**[*]
Google Research
tvmarinov@google.com

**Julian Zimmert**
Google Research
zimmert@google.com

## Abstract

Recent progress in model selection raises the question of the fundamental limits of these techniques. Under specific scrutiny has been model selection for general contextual bandits with nested policy classes, resulting in a COLT2020 open problem. It asks whether it is possible to obtain simultaneously the optimal single algorithm guarantees over all policies in a nested sequence of policy classes, or if otherwise this is possible for a trade-off $\alpha \in [\frac{1}{2}, 1)$ between complexity term and time: $\ln(|\Pi_m|)^{1-\alpha}T^\alpha$. We give a disappointing answer to this question. Even in the purely stochastic regime, the desired results are unobtainable. We present a Pareto frontier of up to logarithmic factors matching upper and lower bounds, thereby proving that an increase in the complexity term $\ln(|\Pi_m|)$ independent of $T$ is unavoidable for general policy classes. As a side result, we also resolve a COLT2016 open problem concerning second-order bounds in full-information games.

## 1 Introduction

Contextual multi-armed bandits are a fundamental problem in online learning [Auer et al., 2002, Langford and Zhang, 2007, Chu et al., 2011, Abbasi-Yadkori et al., 2011]. The contextual bandit problem proceeds as a repeated game between a learner and an adversary. At every round of the game the adversary prepares a pair of a context and a loss over an action space, the learner observes the context and selects an action from the action space and then observes only the loss of the selected action. The goal of the learner is to minimize their cumulative loss. The performance measure, known as *regret*, is the difference between the learner's cumulative loss and the smallest loss of a fixed policy, belonging to an apriori determined policy class, mapping contexts to actions. Given a single contextual bandit instance with finite sized policy class, the well-known Exp4 algorithm [Auer et al., 2002] achieves the optimal regret bound of $\mathcal{O}(\sqrt{KT\ln(|\Pi|)})$. Regret guarantees degrade with the complexity of the policy class, therefore a a learner might want to leverage "guesses" about the optimal policy. Given policy classes $\Pi_1 \subset \cdots \subset \Pi_M$, a learner would ideally suffer regret scaling only with the complexity of $\Pi_{m^*}$, the smallest policy class containing the optimal policy $\pi^*$. While these kind of results are obtainable in full-information games, in which the learner gets to observe the loss for all actions, they are impossible for multi-armed bandits [Lattimore, 2015]. In some aspects, contextual bandits are an intermediate setting between full-information and multi-armed bandits and it is unknown if model selection is possible. Foster et al. [2020b] stated model selection in contextual bandits as a relevant open problem in COLT2020. Any positive result for model selection in contextual bandits would imply a general way to treat multi-armed bandits with a switching baseline. Furthermore any negative result is conjectured to implicate negative results on another unresolved open problem on second order bounds for full-information games [Freund, 2016].

In this paper, we give a fairly complete answer to the questions above.

---

[*]Author was at Johns Hopkins University during part of this work.

P.1 We provide a Pareto frontier of upper bounds for model selection in contextual bandits with finite sized policy classes.

P.2 We present matching lower bounds that shows that our upper bounds are tight, thereby resolve the motivating open problems [Foster et al., 2020b].

P.3 We present a novel impossibility result for adapting to the number of switch points under adaptive adversaries [Besbes et al., 2014].

P.4 We negatively resolve an open problem on second order bounds for full-information [Freund, 2016].

**Related work.** A problem closely related to contextual bandits with finite policy classes are linear contextual bandits. Model selection in linear contextual bandit problems has recently received significant attention, however none of these resuls transfer to the finite policy case. In the linear bandits problem the $m$-th policy class is a subset of $\mathbb{R}^{d_m}$ and the losses $\ell_{t,\pi(x)}, \pi \in \Pi_m$ are linear, that is $\ell_{t,\pi(x)} = \langle \theta_m, \phi_m(x, \pi(x)) \rangle + \xi$. Here $\phi_m : \mathcal{X} \times \mathcal{A} \to \mathbb{R}^{d_m}$ is a feature embedding mapping from context-action pairs into $\mathbb{R}^{d_m}$, $\xi$ is mean-zero sub-Gaussian noise with variance proxy equal to one and $\theta_m \in \mathbb{R}^{d_m}$ is an unknown parameter.

Foster et al. [2019] assume the contexts are also drawn from an unknown distribution $x \sim \mathcal{D}$ and propose an algorithm which does not incur more than $\tilde{O}(\frac{1}{\gamma^3}(i^*T)^{2/3}(Md_{i^*})^{1/3})$, where $\gamma^3$ is the smallest eigenvalue of the covariance matrix of feature embeddings $\Sigma = \mathbb{E}_{x \sim \mathcal{D}} \left[ \frac{1}{M} \sum_{a \in \mathcal{A}} \phi_M(x, a)\phi_M(x, a)^\top \right]$. Pacchiano et al. [2020b] propose a different approach based on the corralling algorithm of Agarwal et al. [2017] which enjoys a $\tilde{O}(d_{i^*}\sqrt{T})$ regret bound for finite action sets and $\tilde{O}(d_{i^*}^2\sqrt{T})$ bound for arbitrary action sets $\mathcal{A}$. Later, Pacchiano et al. [2020a] design an algorithm which enjoys a gap-dependent guarantee under the assumption that all of the miss-specified models have regret $R_i(t) \geq \Delta t, \forall t \in [T]$. Under such an assumption, the authors recover a regret bounds of the order $\tilde{O}(d_{i^*}\sqrt{T} + d_{i^*}^4/\Delta)$ for arbitrary action sets. Cutkosky et al. [2020] also manage to recover the $O(d_{i^*}\sqrt{T})$ and $O(d_{i^*}^2\sqrt{T})$ bounds for the model selection problems through their corralling algorithm. Ghosh et al. [2021] propose an algorithm which enjoys $\tilde{O}\left( \frac{d_M^2}{\gamma^{4.65}} + \sqrt{d_{m^*}T} \right)$ in the finite arm setting, where $\gamma = \min\{|\theta_{m^*,i}| : |\theta_{m^*,i}| > 0\}$ is the smallest, in absolute value, entry of $\theta_{m^*}$. Their algorithm also enjoys a similar guarantee for arbitrary action sets with $\sqrt{d_{m^*}T}$ replaced by $d_{m^*}\sqrt{T}$. Zhu and Nowak [2021] show that it is impossible to achieve the desired regret guarantees of $\sqrt{d_{m^*}T}$ without additional assumptions by showing a result similar to the one of Lattimore [2015]. The work of Lattimore [2015] states that in the stochastic multi-armed bandit problem it is impossible to achieve $\sqrt{T}$ regret to a fixed arm, without suffering at least $K\sqrt{T}$ regret to a different arm.

Chatterji et al. [2020] study the problem of selecting between an algorithm for the linear contextual bandit problem and the simple stochastic multi-armed bandit problem, that is they aim to achieve simultaneously a regret guarantee which is instance-dependent optimal for the stochastic multi-armed bandit problem and optimal for the finite arm stochastic linear bandit problem. The proposed results only hold under additional assumptions. More generally, the study of the corralling problem, in which we are presented with multiple bandit algorithms and would like to perform as well as the best one, was initiated by Agarwal et al. [2017]. Other works which fall into the corralling framework are that of Foster et al. [2020a] who study the miss-specified linear contextual bandit problem, that is the observed losses are linear up to some unknown $\epsilon$ miss-specification, and the work of Arora et al. [2021] who study the corralling problem for multi-armed stochastic bandit algorithms.

Our work also shows an impossibility result for the stochastic bandit problem with non-stationary rewards. Auer [2002] first investigates the problem under the assumption that there are $L$ distributional changes throughout the game and gives an algorithm with a $\tilde{O}(\sqrt{KLT})$ *dynamic regret*[2] bound, under the assumption that $L$ is known. Auer et al. [2019] achieves similar regret guarantees without assuming that the number if switches (or changes) of the distribution is known. A different measurement of switches is the total variation of changes in distribution $V_T = \sum_{t=2}^{T} \|\mathbb{E}[\ell_t] - \mathbb{E}[\ell_{t-1}]\|_\infty$. Multiple works give dynamic regret bounds of the order $\tilde{O}(V_T^{1/3}T^{2/3})$ (hiding dependence on the size of the policy class) when $V_T$ is known, including for extensions of the multi-armed bandit problem like

---

[2]In dynamic regret the comparator is the best action for the current distribution.

| General CB | Upper bound | Lower bound |
|---|---|---|
| **adaptive adversary** | $\mathcal{O}(\max\{\mathfrak{C}, \frac{\ln|\Pi_m|}{\mathfrak{C}}\}\sqrt{MKT})$ | $\Omega(\max\{\mathfrak{C}, \frac{\ln|\Pi_m|}{\mathfrak{C}}\}\sqrt{\frac{KT}{\ln(K)}})$ |
| **oblivious adversary / stochastic** | $\mathcal{O}(\max\{\mathfrak{C}, \frac{\ln|\Pi_m|}{\mathfrak{C}}\}\sqrt{MKT})$ | $\Omega(\max\{\mathfrak{C}, \frac{\ln|\Pi_m|}{\mathfrak{C}}\}\sqrt{T})$ |

| S-switch | Upper bound | Lower bound |
|---|---|---|
| **adaptive adversary** | $\tilde{\mathcal{O}}(\max\{\mathfrak{C}, \frac{S}{\mathfrak{C}}\}\sqrt{KT})$ | $\Omega(\max\{\mathfrak{C}, \frac{S}{\mathfrak{C}}\}\sqrt{KT})$ |
| oblivious adversary | $\tilde{\mathcal{O}}(\sqrt{SKT}+T^{3/4})$ | $\Omega(\sqrt{SKT})$ |
| stochastic | $\tilde{\mathcal{O}}(\sqrt{SKT})$ | $\Omega(\sqrt{SKT})$ |

Table 1: Overview of our results. Our novel contributions are in bold; lower bounds only hold if the expressions are not exceeding $\Theta(T)$. The stochastic/oblivious lower bounds hold only for proper algorithms.

contextual bandits and linear contextual bandits [Besbes et al., 2014, Luo et al., 2018, Besbes et al., 2015, Wei et al., 2017]. Cheung et al. [2019], Zhao et al. [2020] further show algorithms which enjoy a parameter free regret bound of the order $\tilde{O}(V_T^{1/4}T^{3/4})$ (hiding dependence on dimensionality) for the linear bandits problem. The lower bound in Table 1 might seem to contradict such results. In Section 5.1 we carefully explain why this is not the case.

Finally, our lower bounds apply to the problem of devising an algorithm which simultaneously enjoys a second order bound over any fraction of experts. Cesa-Bianchi et al. [2007] first investigate the problem of second order bounds for the experts problem, in which the proposed algorithm maintains a distribution $p_t$ over the set of $K$ experts, during every round of the game. The experts are assumed to have stochastic losses $\ell_t$ and the work shows an algorithm with $\tilde{O}(\sqrt{\sum_{t=1}^{T}\mathbb{V}_{i\sim p_t}[\ell_{t,i}]\log K})$ regret guarantee. Chaudhuri et al. [2009], Chernov and Vovk [2010], Luo and Schapire [2015], Koolen and Van Erven [2015] study a different experts problem in which the comparator class for the regret changes from the best expert in hindsight to the uniform distribution over the best $\lfloor \epsilon K \rfloor$ experts for an arbitrary positive $\epsilon$. The above works propose algorithms which achieve a $\tilde{O}(\sqrt{T\log(1/\epsilon)})$ regret bound for all $\epsilon$ simultaneously. Freund [2016] asks if there exists an algorithm which enjoys both guarantees at the same time, that is, does there exist an algorithm with regret bound $\tilde{O}(\sqrt{\sum_{t=1}^{T}\mathbb{V}_{i\sim p_t}[\ell_{t,i}]\log(1/\epsilon)})$ which holds simultaneously for all positive $\epsilon$.

**Notation.** For any $N \in \mathbb{N}$, $[N]$ denotes the set $\{1, \dots, N\}$. $\tilde{O}$ notation hides poly-logarithmic factors in the horizon $T$ and the number of arms $K$ but not in the size of the policy classes $|\Pi_m|$.

## 2 Problem setting

We consider the contextual bandit problem with general policy classes of finite size. There are $K$ arms and nested policy classes $(\Pi_m)_{m=1}^{M}$, where a policy $\pi \in \Pi_m$, $\pi : \mathcal{X} \to [K]$ is a mapping from an arbitrary context space $\mathcal{X}$ into the set of $K$ arms. The game is played for $T$ rounds and at any time $t$, the agent observes a context $x_t \in \mathcal{X}$, selects arm $A_t \in [K]$ and observes the loss $\ell_{t,A_t}$ from an otherwise unobserved loss vector $\ell_t \in [K]$. We measure an algorithm's performance in terms of pseudo-regret, which is the expected cumulative regret of the player against following a fixed policy in hindsight

$$\text{Reg}(T, \Pi) = \max_{\pi \in \Pi} \mathbb{E}\left[\sum_{t=1}^{T} \ell_{t,A_t} - \ell_{t,\pi(x_t)}\right].$$

**Environments.** We distinguish between *stochastic* environments and *oblivious* or *adaptive* adversaries. In stochastic environments, there are unknown distribution $P_{\mathcal{X}}, Q$ such that $x_t \sim P_{\mathcal{X}}$ and

$\ell_t \sim Q(\cdot|x_t)$ are i.i.d. samples. In the adversarial regime, the distributions can change over time, i.e. $x_t \sim P_{\mathcal{X},t}, \ell_t \sim Q_t(\cdot|x_t)$. When the choices are fixed at the beginning of the game, the adversary is called oblivious, while an adaptive adversary can chose $P_{\mathcal{X},t}, Q_t$ based on all observations up to time $t-1$.

Often the stochastic-adversarial hybrid problem has been studied with adversarially chosen context but stochastic losses. In our work, all upper bounds hold in the stronger notion where both the losses and the contexts are adaptive, while the lower bounds hold for the weaker notion where only the contexts are adaptive.

**Open problem [Foster et al., 2020b].**   The regret upper bounds for all regimes introduced above for a fixed policy class $\Pi$ of finite size are of the order $\tilde{O}(\sqrt{\ln(|\Pi|)KT})$ and can be achieved by the Exp4 algorithm [Auer et al., 2002]. The question asked by Foster et al. [2020b]: For a nested sequence of policies $\Pi_1 \subset \Pi_2 \subset \cdots \subset \Pi_M$, is there a universal $\alpha \in [\frac{1}{2}, 1)$ such that a regret bound of

$$\text{Reg}(T, \Pi_m) = \text{PolyLog}(K, M)\tilde{\mathcal{O}}\left(\ln(|\Pi_m|)^{1-\alpha}T^\alpha\right) \tag{1}$$

is obtainable for all $m \in [M]$ simultaneously?

W.l.o.g. we can assume that $M = \mathcal{O}(\ln\ln(|\Pi_M|)) = \mathcal{O}(\ln(T))$. Otherwise we take a subset of policy classes that includes $\Pi_M$ and where two consequent policy classes at least square in size. Due to nestedness, any guarantees on this subset of models imply up to constants the same bounds on the full set.

**S-switch**   A motivating example for studying nested policy classes is the S-switch problem. The context is simply $x_t = t \in [T]$ and the set of policies is given by

$$\Pi_S = \left\{ \pi \,\middle|\, \sum_{t=1}^{T-1} \mathbb{I}\{\pi_t \neq \pi_{t+1}\} \leq S \right\},$$

the set of policies that changes its action not more than $S$ many times. Any positive result for contextual bandits with finite sized policy classes would provide algorithms that adapt to the number of switch points, since $\ln|\Pi_S| = \tilde{\mathcal{O}}(S)$. To make clear what problem we are considering, we are using $\text{Reg}_{SW}(T, S)$ to denote the regret in the switching problem.

Next, we define the class of *proper* algorithms which choose their policy at every time step $t$ independently of context $x_t$. Restricting our attention to such algorithms greatly reduces the technicalities for lower bound proofs in the non-adaptive regimes. The lower bound for this class of algorithms is also at the core of the argument for adaptive (improper) algorithms in stochastic environments.

**Definition 1.** *We call an algorithm* proper, *if at any time $t$, the algorithm follows the recommendation of a policy $\pi_{i_t} \in \Pi_M$, and if the choice of $i_t$ by the algorithm, is independent of the context $x_t$.*

**Example.**   EXP4 is proper.

The properness assumption intuitively allows us to reduce the model selection problem to a bandit-like problem in the space of all policies $\Pi_M$. We give more details in Section 4.2 and Appendix B.3.

## 3   Upper bounds

In this section, we generalize the Hedged-FTRL algorithm [Foster et al., 2020a] to obtain an upper bound for model selection over a large collection of $\sqrt{T}$ regret algorithms.

**Theorem 1.** *For any $\mathfrak{C} > 0$, we can tune Hedged-FTRL over a selection of $M$ instances of EXP4 operating on policy classes $\Pi_1, \ldots \Pi_M$, such that the following regret bound holds uniformly over all $m \in [M]$*

$$\text{Reg}(T, \Pi_m) = \tilde{\mathcal{O}}\left( \max\left\{ \mathfrak{C}, \frac{\ln|\Pi_m|}{\mathfrak{C}} \right\} \sqrt{MKT} \right).$$

**Hedged-FTRL.** $(\alpha, R)$-hedged FTRL, introduced in Foster et al. [2020a], is a type of Follow the Regularized Leader (FTRL) algorithm which is used as a corralling algorithm [Agarwal et al., 2017]. At every round $t$, the algorithm chooses to play one of $M$ base algorithms $(Base_i)_{i=1}^M$. Base algorithm $i$ is selected with probability $q_{t,i}$, where $q_t \in \Delta^{M-1}$ is a distribution over base algorithms determined by the FTRL rule $q_t = \arg\min_{q \in \Delta^{M-1}} \langle q, L_t - B_t \rangle + F(q)/\eta$, where $L_t \in \mathbb{R}_+^M$ is the sum of the loss vectors $(\mathbf{e}_{M_s} \ell_{s,A_s}/q_{s,M_s})_{s=1}^{t-1}$, $F : \Delta^{M-1} \to \mathbb{R}$ is the potential induced by the $\alpha$-Tsallis entropy, $\eta$ is a

**Input:** $\alpha, R, \beta, Top, (Base_i)_{i=1}^M$
**for** $t = 1, \ldots, T$ **do**
  Get $M_t, q_{t,M_t}$ from $Top$
  Let $Base_{M_t}$ play the next round and receive $A_t$
  Play $A_t$ and observe $\ell_{t,A_t}$
  Update $Base_{M_t}$ with $\ell_{t,A_t}/q_{t,M_t}$
  Update $Top$ with $(M_t, \ell_t)$
  **if** $q_{t+1}$ *would violate Eq.* (2) **then**
    $\quad$ Bias losses by $b_t$ to ensure Eq. (2).
  **end**
**end**

**Algorithm 1:** Hedged FTRL

step size determined by the problem parameters and $B_t$ is a special bias term which we now explain. Define $\rho_{t,m}^{-1} = \min\{\beta_m, \min_{s \in [t]} q_{s,m}\}$, and initialize $B_{0,m} = \rho_{1,m}^\alpha R_m$. Here $\rho_t$ is a vector which tracks the variance of the loss estimators, $R$ is a vector with regret upper bounds for the base algorithms, and $\beta \leq q_1$ is a threshold depending on the base algorithms. At any time $t$, after selecting base algorithm $M_t$ to play the current action, the top (corralling) algorithm observes its loss and gives as feedback an important weighted loss to the selected base, $M_t$. Whenever the base played at round $t$ would satisfy $\rho_{t+1,M_t} > \rho_{t,M_t}$, the loss fed to the top algorithm is adjusted with a bias $b_{t,M_t}$, such that the cumulative biases track the quantity $\rho_{t+1,m}^\alpha R_m$. This has been shown to be always possible [Foster et al., 2020a]. The condition for adjusting the biases reads

$$\forall m \in [M] : \; B_{0,m} + \sum_{s=1}^t b_{s,m} = \rho_{t+1,m}^\alpha R_m \,. \tag{2}$$

The condition in Equation 2 is motivated in a similar way to the stability condition in the work of Agarwal et al. [2017]. Algorithm 1 constructs an unbiased estimator for the loss vector, $\mathbf{e}_{M_t} \ell_{t,A_t}/q_{t,M_t}$, and updates each base algorithm accordingly. A similar update is present in the CORRAL algorithm Agarwal et al. [2017] in which each of the base learners also receives an importance weighted loss. The regret of the base learners is assumed to scale with the variance of the importance weighted losses. This assumption is natural and in practice holds for all bandit or expert algorithms. The scaling of the regret, however, must be appropriately bounded as Agarwal et al. [2017] show, otherwise no corralling or model selection guarantees are possible. Formally, the following stability property is required. If an algorithm $\mathcal{B}$ enjoys a regret bound $R$ under environment $\mathcal{V}$ with loss sequence $(\ell_t)_{t=1}^T$, then the algorithm is $(\alpha, R)$-*stable* if it enjoys a regret bound of the order $\mathbb{E}[\rho_{\max}^\alpha]R$ under the environment $\mathcal{V}'$ of importance weighted losses $(\hat{\ell}_t)_{t=1}^T$, where $\rho_{\max}$ is the maximum variance of the $T$ losses and the expectation is taken with respect to any randomness in $\mathcal{B}$. Essentially all bandit and expert algorithms used in practice are $(\alpha, R)$-stable with $\alpha \leq 1/2$, e.g., Exp4 is $(1/2, \sqrt{KT \ln(|\Pi|)})$-stable. The bias terms in Algorithm 1 intuitively cancel the additional variance introduced by the importance weighted losses and this is why we require the biases to satisfy Equation 2.

**Theorem 2.** *Given a collection of base algorithms* $(\mathcal{B}_m)_{m=1}^M$ *which are* $(1/2, \sqrt{\mathfrak{C}_m T})$-*stable, that is*

$$\forall m \in [M] : \; \mathrm{Reg}_{\mathrm{Imp}}(T, \mathcal{B}_m) \leq \mathbb{E}[\sqrt{\rho_{Tm}}]\sqrt{\mathfrak{C}_m T} \,,$$

*and any* $\mathfrak{C} \geq 0$, *then the regret of* $(1/2, R, \beta)$-*hedged Tsallis-Inf with* $R_m = \sqrt{\mathfrak{C}_m T}$, $\beta_m = \frac{1}{M} \max\{1, \frac{\mathfrak{C}^2}{\mathfrak{C}_m}\}$ *satisfies a simultaneous regret of*

$$\forall m \in [M] : \; \mathrm{Reg}(T, \mathcal{B}_m) \leq 2 \max\left\{\mathfrak{C}, \frac{\mathfrak{C}_m}{\mathfrak{C}}\right\} \sqrt{MT} + \sqrt{2MT} \,.$$

The analysis follows closely the proof of Foster et al. [2020a] and is postponed to Appendix A.

Theorem 2 recovers the bounds of Pacchiano et al. [2020b] for model selection in linear bandits, but holds in more general settings including adaptive adversaries in both contexts and losses. It neither requires nestedness of the policies nor that the policies operate on the same action or context space.

*Proof of Theorem 1.* The EXP4 algorithm initialized with policy class $\Pi_m$ satisfies the condition of Theorem 2 with $\mathfrak{C}_m = \mathcal{O}(\ln|\Pi_m|)$, as shown in Agarwal et al. [2017]. Hence Theorem 1 is a direct corollary of Theorem 2. $\qquad \square$

## 4 Lower bounds

We present lower bounds that match the upper bounds from Section 3 up to logarithmic factors, thereby proving a tight Pareto frontier of worst-case regret guarantees in model selection for contextual bandits.

In the first part of this section, we consider a special instance of $S$-switch with *adaptive* adversary. The proof technique based on Pinsker's inequality is folklore and leads to the following theorem.

**Theorem 3.** *For any $K \geq 3$, sufficiently large $T$, and any algorithm with regret guarantee*

$$\mathrm{Reg}_{SW}(T, 1) = \mathcal{O}(\mathfrak{C}\sqrt{KT}),$$

*there exists for any number of switches $S = \Omega(\mathfrak{C}^2)$ a stochastic bandit problem such that*

$$\mathrm{Reg}_{SW}(T, S) = \Omega\left(\min\left\{\frac{S}{\mathfrak{C}}\sqrt{KT}, T\right\}\right).$$

*This bound holds even when the agent is informed about the number of switches up to time $t$.*

Since this bound holds even when the agent is informed about when a switch occurs, we can restrict the policy class to policies that only switch arms whenever the agent is informed about a switch in the environment. This as a contextual bandit problem with context $\mathcal{X} = [S+1]$ and $|\Pi_S| = \Theta(K^S)$ policies. Hence Theorem 3 implies a lower bound of $\mathrm{Reg}(T, \Pi_S) = \Omega\left(\min\left\{\frac{\ln|\Pi_S|}{\mathfrak{C}\ln(K)}\sqrt{KT}, T\right\}\right)$.

In the second part of the section, we consider the *stochastic* regime. Our lower bound construction is non-standard and relies on bounding the total variation between problem instances directly without the use of Pinsker's inequality.

**Theorem 4.** *There exist policy classes $\Pi_1 \subset \Pi_2$ [3] with $|\Pi_2| = \Omega(\mathfrak{C}^2)$, such that if the regret of a proper algorithm is upper bounded in any environment by*

$$\mathrm{Reg}(T, \Pi_1) = \mathcal{O}(\mathfrak{C}\sqrt{T}),$$

*then there exists an environment such that*

$$\mathrm{Reg}(T, \Pi_2) = \Omega\left(\max\left\{\mathfrak{C}, \frac{\ln|\Pi_2|}{\mathfrak{C}}\right\}\sqrt{T}\right).$$

These theorems directly provide negative answers to [Foster et al., 2020b].

**Corollary 1.** *There is no $\alpha \in [\frac{1}{2}, 1)$ that satisfies the regret guarantee of open problem (1) for any algorithm in the adaptive adversarial regime or any proper algorithm in the stochastic case.*

*Proof.* By Theorems 3 and 4, for any $\alpha > 0$ there exists $K = 3$, $M = 2$, $|\Pi_1| = 1$, $|\Pi_2| = \Theta(\exp(T^\alpha))$. Assume that $\mathrm{Reg}(T, \Pi_1) \leq C_T T^\alpha = C_T T^{\alpha - \frac{1}{2}}\sqrt{T}$, where $C_T = \mathrm{PolyLog}(T)$. Hence by Theorem 3 and Theorem 4 there exist environments where

$$\mathrm{Reg}(T, \Pi_2) = \Omega\left(\frac{T^\alpha}{C_T T^{\alpha - \frac{1}{2}}}\sqrt{T}\right) = \tilde{\Omega}(T).$$

$\square$

Finally, we disprove the open problem in the stochastic case for any algorithm.

**Theorem 5.** *No algorithm (proper or improper) can satisfy the requirements of open problem (1) for all stochastic environments.*

We present the high level proof ideas in the following subsections and the detailed proof in Appendix B.

---

[3]In Foster et al. [2020b] open problem 2, they ask about model based contextual bandit with realizability. Our lower bound is providing an instance of that.

## 4.1 Adaptive adversary: $S$-switch($\Delta$) Problem

We present the adaptive environment in which model selection fails and the proof of Theorem 3.

The adversary switches the reward distribution up to $S$ many times, thereby segmenting the time into $S + 1$ phases $(1, \ldots, \tau_1, \tau_1 + 1, \ldots, \tau_2, \ldots, \tau_S, \ldots T)$. We denote $x_t \in [S + 1]$ as the counter of phases and assume the agent is given this information. For each phase $x_t \in [S]$, the adversary selects an optimal arm $(a_s^*)_{s=1}^S$ uniformly at random among the first $K - 1$ arms. If $x_t \leq S$, the losses are i.i.d. Bernoulli random variables with means

$$\mathbb{E}[\ell_{t,i}] = \frac{1}{2} - \begin{cases} 0 & \text{for } i \in [K-1] \setminus \{a_{x_t}^*\} \\ \Delta & \text{for } i = a_{x_t}^* \\ \frac{7}{8}\Delta & \text{for } i = K. \end{cases}$$

In phase $S + 1$, all losses are 0 until the end of the game. The adversary decides on the switching points based on an adaptive strategy. A switch from phase $s < S + 1$ to $s + 1$ occurs when the player has played $\mathrm{N}_{\max} = \lceil \frac{K-1}{192\Delta^2} \rceil$ times an arm in $[K - 1]$ in phase $s$. We can see this problem either as a special case of S-switch problem, or alternatively as a contextual bandit problem with $|\Pi_S| = K^{S+1}$ policies.

The lower bound proof for $S$-switch($\Delta$) relies on the following Lemma, which is proven in Appendix B.

**Lemma 1.** *Let an agent interact with a $K - 1 \geq 2$ armed bandit problem with centered Bernoulli losses and randomized best arm of gap $\Delta \leq \frac{1}{8\sqrt{3}}$ for an adaptive number of time steps $N$. If the probability of $N \geq \mathrm{N}_{\max} = \lceil \frac{K-1}{192\Delta^2} \rceil$ is at least $\frac{1}{2}$, then the regret after $\mathrm{N}_{\max}$ time-steps conditioned on the event $N \geq \mathrm{N}_{\max}$ is lower bounded by*

$$\mathrm{Reg} \geq \frac{\Delta}{4} \mathrm{N}_{\max}.$$

Informally, this Lemma says that conditioned on transitioning from phase $s$ to phase $s + 1$, the agent has suffered regret $\Omega(\Delta \, \mathrm{N}_{\max})$ against arm $K$ during phase $s$.

*Informal proof of Theorem 3.* The adversary's strategy is designed in a way such that at each phase $s \in [S]$ it only allows the player's strategy to interact with the environment just enough times to discover the best action $a_s^*$. Then a new phase begins to prevent the player from exploiting knowledge of $a_s^*$. This ensures by Lemma 1 that the player suffers regret at least $\Omega(\Delta N_{\max})$ during each completed phase. If an agent proceeds finding $a_s^*$ for all phases $s \in [S]$, then the regret against the non-switching baseline is $\mathrm{Reg}_{SW}(T, 1) = O(\Delta \, \mathrm{N}_{\max} \, S)$. By the assumption on the maximum regret of $\mathrm{Reg}_{SW}(T, 1)$ and an appropriate choice of $N_{\max}$ and $\Delta$, we can ensure that the agent must fail to discover all $a_s^*$ with constant probability, thus incurs regret at least $\mathrm{Reg}_{SW}(T, S) = \Omega(\Delta T)$ against the optimal $S$-switch baseline. Tuning $\Delta$ and $\mathrm{N}_{\max}$ yield the desired theorem. The formal argument with explicit choice of $\Delta$ is found in Appendix B. $\square$

## 4.2 Stochastic lower bound

We now present the stochastic environment used for the impossibility results in Theorems 4 and 5.

There are $k + 1$ environments $(\mathcal{E}_i)_{i=0}^k$ with $\Pi_2 = \{\pi_i | i \in [k] \cup \{0\}\}$ policies and $\Pi_1 = \{\pi_0\}$. In all environments, we have $K = 3$ and $\pi_0$ always chooses action 3, while $(\pi_i)_{i=1}^k$ are playing an action from $\{1, 2\}$ uniformly at random. (In other words, the context is $\mathcal{X} = \{1, 2\}^k$ with $x_t$ sampled uniformly at random and $\pi_i(x) = x_i$.)

In each environment, the losses of actions $\{1, 2\}$ at any time step satisfy $\ell_{t,1} = 1 - \ell_{t,2}$, which are conditioned on $x_t$ independent Bernoulli random variables, with mean

$$\mathbb{E}_{\mathcal{E}_0}[\ell_{t,1}] = \mathbb{E}_{\mathcal{E}_0}[\ell_{t,2}] = \frac{1}{2} \qquad \text{and } \forall i \in [k] : \mathbb{E}_{\mathcal{E}_i}[\ell_{t,\pi_i(x_t)}] = \frac{1}{2}(1 - \Delta).$$

Action 3 gives a constant loss of $\frac{1}{2} - \frac{1}{4}\Delta$ in all environments.

Let us unwrap these definitions. Playing either action 1 or action 2, which we call *revealing* actions, yields *full-information* of all random variables at time $t$ due to the dependence of $\ell_{t,1} = 1 - \ell_{t,2}$ and

the non-randomness of $\ell_{t,3}$. On the other hand, playing action 3 allows only to observe $x_t$, which has the same distribution in all environments, hence there is no information gained at all.

We know from full-information lower bounds that for optimal tuning of the gap, one suffers $\Omega(\sqrt{\ln(k)T})$ regret in the policy class $\Pi_2$, due to the difficulty of identifying the optimal arm. For a smaller regret in policy class $\Pi_1$, one needs to confirm or reject the hypothesis $\mathcal{E}_0$ faster than it takes to identify the optimal arm. Existing techniques do not answer the question whether this is possible, and our main contribution of this section is to show that the hardness of rejecting $\mathcal{E}_0$ is of the same order as identifying the exact environment.

For the remaining section, it will be useful to consider a reparametrization of the random variables. Let $z_t \in \{0,1\}^k$ be the losses incurred by the policies $(\pi_i)_{i=1}^k$: $z_{t,i} = \ell_{t,\pi_i(x_t)}$. We can easily see that $z_t$ together with $x_{t,1}$ is sufficient to uniquely determine $\ell_t$ and $x_t$. Furthermore, $z_t$ is always a vector of independent Bernoulli random variables, which are independent of $x_{t,1}$[4]. In environments $(\mathcal{E}_i)_{i=1}^k$, the $i$-th component is a biased Bernoulli, while all other components have mean $\frac{1}{2}$. In $\mathcal{E}_0$, no component is biased. As before, $x_{t,1}$ does not provide any information since its distribution conditioned on $z_t$ is identical in all environments (see Lemma 4 in Appendix B for a formal proof).

Under this reparameterization and ignoring non-informative bits of randomness, the problem of distinguishing $\mathcal{E}_0$ from $\{\mathcal{E}_i\}_{i=1}^k$ now looks as follows. For time steps $t = 1, \ldots, T$, decide whether to play a revealing action and observe $z_t$ (potentially by taking $x_t$ into account). Use observed $(z_{\tau_n})_{n=1}^N$ to distinguish between the environments. *Proper* algorithms simplify the problem even further, because selecting $\pi_{i_t}$ independently of $x_t$ implies that the decision of observing $z_t$ is also independent of $x_t$ (any policy except $\pi_0$ allows to observe $z_t$ under any context). Hence for proper algorithms, we can reason directly about how many samples $z_t$ are required to distinguish between environments. This problem bears similarity to the property testing of dictator functions [Balcan et al., 2012] and sparse linear regression [Ingster et al., 2010][5], however, there is no clear way to apply such results to our setting.

The following lemma shows the difficulty of testing for hypothesis $\mathcal{E}_0$.

**Lemma 2.** *Let $\Delta \leq \frac{1}{4}$, $k \geq e^{20} + 1$, $N \leq \lfloor \frac{\ln(k-1)}{20\Delta^2} \rfloor$ and $\Delta^2 N \geq \frac{1}{2}$. If the algorithm chooses whether to reveal $z_t$ independently of $x_t$ and if the total times $z_t$ is revealed is bounded by $N$ a.s. then for any measurable event $E$ it holds that*

$$\min_{i \in [k]} \mathbb{P}_{\mathcal{E}_i}(E) - \mathbb{P}_{\mathcal{E}_0}(E) \leq \frac{17}{\sqrt[4]{k-1}} \leq \frac{1}{4}.$$

The proof of Lemma 2 is deferred to Appendix B.3. The high level idea is to directly bound the TV between $\min_{i \in [k]} \mathbb{P}_{\mathcal{E}_i}$ and $\mathbb{P}_{\mathcal{E}_0}$ over the space of outcomes of $(z_{t_n})_{n=1}^N$ by utilizing Berry-Essen's inequality instead of going through Pinsker's inequality. This step is key to achieve a dependence on $k$ in the bound.

For readers familiar with lower bound proofs for bandits and full-information, this Lemma should not come at a huge surprise. For a $T$-round full-information game, it tells us that we can bias a single arm up to $\Delta = \Omega(\sqrt{\ln(k)/T})$, without this being detectable. This directly recovers the well known lower bound of $\Delta T = \Omega(\sqrt{\ln(k)T})$ for full-information via the argument used for bandit lower bounds. However, this result goes beyond what is known in the literature. We not only show that one cannot reliably detect the biased arm, but that one cannot even reliably detect whether any biased arm is present at all. This property is the key to showing the lower bound of Theorem 4.

*Informal proof of Theorem 4.* Under environment $\mathcal{E}_0$, observing $z_t$ for $n$ time-steps incurs a regret of $\mathrm{Reg}_{\mathcal{E}_0}(T, \Pi_1) = \Omega(\Delta n)$. Using the assumption on the regret $\mathrm{Reg}_{\mathcal{E}_0}(T, \Pi_1)$ and Markov inequality, we obtain an upper bound $N$ on the expected number of observations, which holds with probability $\frac{1}{2}$. We can construct an algorithm $\mathcal{A}$ that never observes more than $N$ samples, by following algorithm $\mathcal{A}$ until it played $N$ times a revealing action and then commits to policy $\pi_0$ (action 3). Since the algorithm $\underline{\mathcal{A}}$ is proper, we can define $Z = (z_{\tau_i})_{i=1}^N$ as the observed $z$'s during time $\tau_i$ where the algorithm plays a revealing action. For the revealed information generated by $\underline{\mathcal{A}}$, we tune the

---

[4]We want to emphasize that $z_t$ is only independent of $x_{t,1}$, not independent of the full vector $x_t$.

[5]The setting of [Ingster et al., 2010] is different from our setting as they consider an asymptotic regime where both feature sparsity and dimensionality of the problem go to infinity, while for us the sparsity is fixed to one.

remaining parameters such that the conditions of Lemma 2 are satisfied. Let $E$ be the event that $\underline{\mathcal{A}}$ plays exactly $N$ times a revealing action (i.e. $\mathcal{A}$ plays at least $N$ time the revealing action), then $E$ happens with probability $1 - \Omega(1)$ under $\min_{i \in [k]} \mathbb{P}_{\mathcal{E}_i}(E)$. Thus, there exists an environment $i \in [k]$ such that $\mathcal{A}$ plays less than $N$ times an action in $\{1, 2\}$ with constant probability, which incurs regret of $\mathrm{Reg}_{\mathcal{E}_i}(T, \Pi_2) = \Omega(\Delta T)$. The theorem follows from tuning $\Delta$ and $N$, which is done formally in Appendix B. $\qquad \square$

**Improper algorithms.** Even though we are not able to extend the lower bound proof uniformly over all values $\mathfrak{C}$ and $k$ to improper algorithms, we can still show that no algorithm (proper or improper) can solve the open problem (1) for stochastic environments.

The key is the following generalization of Lemma 2, which is proven in the appendix.

**Lemma 3.** *Let* $\Delta \le \frac{1}{4}$, $k \ge e^{20} + 1$, $N \le \lfloor \frac{\ln(k-1)}{20\Delta^2} \rfloor$ *and* $\Delta^2 N \ge \frac{1}{2}$. *If the total number of times* $z_t$ *is revealed is bounded by* $N$ *a.s. then for any measurable event* $E$ *it holds that*

$$\min_{i \in [k]} \mathbb{P}_{\mathcal{E}_i}(E) - \mathbb{P}_{\mathcal{E}_0}(E) \le \frac{17 T^N}{\sqrt[4]{k-1}}.$$

*This holds even if the agent can take all contexts* $(x_t)_{t=1}^T$ *and previous observations into account when deciding whether to pick a revealing action at any time-step.*

*Informal proof of Theorem 5.* The proof is analogous to Theorem 4, however we use Lemma 3 to bound the difference in probability of $E$ under $\mathcal{E}_0$ and $\mathcal{E}_i$. The key is to find a tuning such that the RHS of Lemma 3 does not exceed $\frac{1}{4}$. Note that this is of order $\exp(\mathcal{O}(\ln(T)N) - \Omega(\ln(k)))$. Let $\Delta = \Theta(1)$, then the requirement on $\mathrm{Reg}_{\mathcal{E}_0}(T, \Pi_1)$ yields $N = \mathcal{O}(T^\alpha)$. Setting $k = \Theta(\exp(T^{\alpha+\epsilon}))$ for any $\epsilon \in (0, 1 - \alpha)$, then it follows immediately that the RHS in Lemma 3 goes to 0 for $T \to \infty$. Following the same arguments as in Theorem 4, there exists a sufficiently large $T$ up from which there always exists an environment $(\mathcal{E}_i)_{i \in [k]}$ such that the regret is linear in $T$, thereby contradicting the open problem. The formal proof is deferred to Appendix B $\qquad \square$

## 5 Implications

The relevance of open problem Eq. (1) has been motivated by its potential implications for other problems such as the S-switch bandit problem and an unresolved COLT2016 open problem on improved second order bounds for full-information. Our negative result for Eq. (1) indeed lead to the expected insights.

### 5.1 S-switch

Our lower bound in the *adaptive* regime shows that adapting to the number of switches is hopeless if the timing of the switch points is not independent of the players actions. Any algorithm adaptive to the number of switches in the regime with *oblivious* adversary must break in the *adaptive* case, which rules out bandit over bandit approaches based on importance sampling [Agarwal et al., 2017]. The successful algorithm proposed in Cheung et al. [2019] is using a bandit over bandit approach without importance sampling. Nonetheless, all components have adaptive adversarial guarantees. The algorithm splits the time horizon into equal intervals of length $L$. It initializes EXP3 with $\ln(T)$ arms, corresponding to a grid of learning rates. For each epoch, the EXP3 top algorithm samples an arm and starts a freshly initialized instance of EXP3.S using the learning rate corresponding to the selected arm. This instance is run over the full epoch of length $L$. It collects the accumulated losses $L_{sum} = \sum_{t=1}^L \ell_t$ of the algorithm and feeds the loss $L_{sum}/L$ to the EXP3 top algorithm.

If all algorithms in the protocol enjoy guarantees against adaptive adversaries, why do bandit over bandit break against adaptive adversaries? Adaptive adversaries are assumed to pick the losses $\ell_t$ independent of the choice of arm $A_t$ of the agent at round $t$. In the bandit over bandit protocol, the loss of the arm of the top algorithm dependents on the losses that the selected base suffers in the epoch. An adaptive adversary can adapt the losses in the epoch based on the actions of the base algorithm, that means the loss $\ell_t$ is not chosen independent of the action $A_t$. Hence the protocol is broken and the adaptive adversarial regret bounds do not hold.

## 5.2 Second order bounds for full information.

In an unresolved COLT2016 open problem, Freund [2016] asks if it is possible to ensure a regret bound of order

$$\text{Reg}_\varepsilon = \tilde{\mathcal{O}}\left(\sqrt{\sum_{t=1}^{T} \mathbb{V}_{i \sim p_t}[\ell_{t,i}] \ln(\frac{1}{\varepsilon})}\right), \tag{3}$$

against the best $\varepsilon$ proportion of policies simultaneously for all $\varepsilon$. We go even a step further and show that the lower bound construction from Section 4.2 directly provides a negative answer for any $\alpha < 1$ to the looser bound

$$\text{Reg}_\varepsilon = \tilde{\mathcal{O}}\left(\sqrt{\sum_{t=1}^{T} (\mathbb{V}_{i \sim p_t}[\ell_{t,i}] + T^\alpha) \ln(\frac{1}{\varepsilon})}\right). \tag{4}$$

**Theorem 6.** *An algorithm satisfying Eq. (4) for $\alpha < 1$ implies the existence of a proper algorithm that violates the lower bound for the counter example in the proof of Theorem 4.*

Theorem 6 has the following interpretation. For any fixed $\alpha \in [0, 1)$, there is no algorithm which enjoys a regret upper bound as in Equation 3 for all problem instances s.t. $\sum_{t=1}^{T} (\mathbb{V}_{i \sim p_t}[\ell_{t,i}]) = \Theta(T^\alpha)$. This implies we can not hope for a polynomial improvement, in terms of time horizon, over the existing bound of $\tilde{O}(\sqrt{T \ln(1/\epsilon)})$. The detailed proof is found in Appendix B. The high level idea is to initialize the full-information algorithm satisfying Eq. (4) with a sufficient number of copies of the baseline policy $\pi_0$ and to feed importance weighted losses of the experts (i.e. policies) to that algorithm.

As we mention in Section 1, the case $\alpha = 1$ is obtainable. Our reduction relates the adaptation to variance to the model selection problem. As in Eq. (1), $\alpha$ is the trade-off between time and complexity. An algorithm satisfying Eq. (4) with $\alpha = 1$ merely allows to recover the trivial $\mathcal{O}(T)$ bound for model selection, and hence does not lead to a contradiction.

## 6 Conclusion

We derived the Pareto Frontier of minimax regret for model selection in Contextual bandits. Our results have resolved several open problems [Foster et al., 2020b, Freund, 2016].

## Acknowledgments and Disclosure of Funding

We like to thank Haipeng Luo and Yoav Freund for discussions about our lower bound proofs. We thank Tor Lattimore for pointing us to the technicalities required for bounding the total variation of improper algorithms.

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
