## A Upper bound proofs

*Proof of Theorem 2.* Denote with $m_t$ the arm that algorithm $m$ would choose if it would be selected during round $t$. We decompose the regret into

$$\text{Reg}(T, \Pi_m) = \max_{\pi \in \Pi_m} \mathbb{E}\left[\sum_{t=1}^{T} \ell_{t,A_t} - \ell_{t,\pi(x_t)}\right]$$

$$= \max_{\pi \in \Pi_m} \mathbb{E}\left[\sum_{t=1}^{T} \ell_{t,A_t} - \ell_{t,m_t} + \ell_{t,m_t} - \ell_{t,\pi(x_t)}\right]$$

$$= \mathbb{E}\left[\sum_{t=1}^{T} \ell_{t,A_t} - \ell_{t,m_t}\right] + \max_{\pi \in \Pi_m} \mathbb{E}\left[\sum_{t=1}^{T} \frac{\mathbb{I}(M_t = m)}{q_{t,m}}(\ell_{t,m_t} - \ell_{t,\pi(x_t)})\right]$$

$$\leq \mathbb{E}\left[\sum_{t=1}^{T} \ell_{t,A_t} - \ell_{t,m_t}\right] + \mathbb{E}[\sqrt{\rho_{T,m}}]\sqrt{\mathfrak{C}_m T},$$

where the last line is by the assumption of the theorem. The first term requires some basic properties of FTRL analysis, see e.g. [Zimmert and Seldin, 2021]. Define $\hat{L}_t = \sum_{s=1}^{t} \hat{\ell}_t$, $\tilde{B}_t = \sum_{s=1}^{t} b_t$. For Tsallis-INF with constant learning rate[6] $\eta = \frac{1}{\sqrt{T}}$, we have the following properties

$$F(q) = -2\sum_{i=1}^{M} \sqrt{q_i}$$

$$\bar{F}^*(-L) = \max_{q \in \Delta([M])} \langle q, -L \rangle - \eta^{-1} F(q)$$

$$q_t = \nabla \bar{F}^*(-(\hat{L}_{t-1} - \tilde{B}_{t-1})).$$

The standard FTRL proof (e.g. Zimmert and Seldin [2021]) shows that

$$\forall t : \mathbb{E}_{M_t \sim q_t}[D_{\bar{F}^*}(-(\hat{L}_t - \tilde{B}_{t-1}), -(\hat{L}_{t-1} - \tilde{B}_{t-1}))] \leq \eta\sqrt{K}.$$

The first term is

$$\mathbb{E}\left[\sum_{t=1}^{T} \ell_{t,A_t} - \ell_{t,m_t}\right] = \mathbb{E}\left[\sum_{t=1}^{T} \langle q_t - \mathbf{e}_m, \hat{\ell}_t \rangle\right]$$

$$= \mathbb{E}\left[\sum_{t=1}^{T} \bar{F}^*(-(\hat{L}_{t-1} - \tilde{B}_{t-1})) - \bar{F}^*(-(\hat{L}_t - \tilde{B}_{t-1}))\right.$$

$$\left. + D_{\bar{F}^*}(-(\hat{L}_t - \tilde{B}_{t-1}), -(\hat{L}_{t-1} - \tilde{B}_{t-1}))\right] - \mathbb{E}\left[\hat{L}_{T,m}\right].$$

Let us consider the terms $\bar{F}^*(-(\hat{L}_{t-1} - \tilde{B}_{t-1}))$ and $\bar{F}^*(-(\hat{L}_t - \tilde{B}_{t-1}))$. First, using the definition of the conjugate function and $q_t$ we know that

$$\bar{F}^*(-(\hat{L}_{t-1} - \tilde{B}_{t-1})) + \bar{F}(q_t) = \langle q_t, -(\hat{L}_{t-1} - \tilde{B}_{t-1}) \rangle.$$

Further by Young's inequality it holds that

$$\bar{F}^*(-(\hat{L}_t - \tilde{B}_{t-1})) + \bar{F}(q_{t+1}) \geq \langle q_{t+1}, -(\hat{L}_t - \tilde{B}_{t-1}) \rangle \implies$$

$$-\bar{F}^*(-(\hat{L}_t - \tilde{B}_{t-1})) \leq \bar{F}(q_{t+1}) + \langle q_{t+1}, \hat{L}_t - \tilde{B}_{t-1} \rangle$$

The above two displays imply

$$\bar{F}^*(-(\hat{L}_t - \tilde{B}_t)) - \bar{F}^*(-(\hat{L}_t - \tilde{B}_{t-1})) \leq \bar{F}(q_{t+1}) + \langle q_{t+1}, \hat{L}_t - \tilde{B}_{t-1} \rangle$$

$$- \bar{F}(q_{t+1}) - \langle q_{t+1}, \hat{L}_t - \tilde{B}_t \rangle$$

$$= \langle q_{t+1}, b_t \rangle.$$

---

[6]The proof can be adapted to time dependent learning rates.

Thus we can bound

$$\sum_{t=1}^{T} \bar{F}^*(-(\hat{L}_{t-1} - \tilde{B}_{t-1})) - \bar{F}^*(-(\hat{L}_t - \tilde{B}_{t-1})) \leq \bar{F}^*(0) - \bar{F}^*(-(\hat{L}_T - B_{T-1})) + \sum_{t=1}^{T-1} \langle q_{t+1}, b_t \rangle$$

$$\leq \bar{F}^*(0) - \bar{F}(\mathbf{e}_m) + \langle \mathbf{e}_m, \hat{L}_T \rangle - \langle \mathbf{e}_m, \tilde{B}_{T-1} \rangle$$

$$+ \sum_{t=1}^{T-1} \langle q_{t+1}, b_t \rangle$$

$$\leq \bar{F}^*(0) - \bar{F}(\mathbf{e}_m) + \langle \mathbf{e}_m, \hat{L}_T \rangle - \sqrt{\rho_{T,m}} R_m$$

$$+ \sqrt{\rho_{1,m}} R_m + \sum_{t=1}^{T-1} \langle q_{t+1}, b_t \rangle$$

$$\leq \frac{\sqrt{M}}{\eta} + \langle \mathbf{e}_m, \hat{L}_T \rangle - \sqrt{\rho_{T,m}} R_m$$

$$+ \sqrt{\rho_{1,m}} R_m + \sum_{t=1}^{T-1} \langle q_{t+1}, b_t \rangle.$$

Taking expectation and setting $\eta$ appropriately we have that

$$\mathbb{E}\left[\sum_{t=1}^{T} \ell_{t,A_t} - \ell_{t,m_t}\right] \leq 2\sqrt{2MT} + \mathbb{E}\left[\sum_{t=1}^{T} \langle q_{t+1}, b_t \rangle\right] - \left(\mathbb{E}[\sqrt{\rho_{T,m}}] - \frac{\sqrt{M\mathfrak{C}_m}}{\mathfrak{C}}\right)\sqrt{\mathfrak{C}_m T}.$$

Finally the final term is

$$\sum_{t=1}^{T} \langle q_{t+1}, b_t \rangle = \sum_{i=1}^{M} \sum_{t=1}^{T} \rho_{t+1,i}^{-1}(\sqrt{\rho_{t+1,i}} - \sqrt{\rho_{t,i}}) R_i$$

$$\leq \sum_{i=1}^{M} \int_{\sqrt{\beta_i}}^{\infty} q^{-1} \, dq \, R_i$$

$$= \sum_{i=1}^{M} \frac{R_i}{\sqrt{\beta_i}} = \mathfrak{C}\sqrt{MT}.$$

Putting all the bounds together finishes the proof. $\qquad\square$

## B  Lower bounds

### B.1  Adaptive adversary lower bound and proof of Theorem 3

*Proof of Lemma 1.* We denote two environments $\mathcal{E}_0, \mathcal{E}_1$. In both environments, we define the random variable $A^* \in [K-1]$ chosen uniformly at random and obliviously to the agents. In the environment $\mathcal{E}_0$, it is not possible to observe any information about $A^*$ because the losses at any time step are sampled i.i.d. $\ell_t \sim \mathcal{B}(\frac{1}{2})$ independent of the action picked. In environment $\mathcal{E}_1$, the loss is still $\ell_t \sim \mathcal{B}(\frac{1}{2})$ when $A_t \neq A^*$, but differs when $A_t = A^*$. In this case, it is instead drawn according to $\ell_t \sim \mathcal{B}(\frac{1}{2} - \Delta)$. The agent might interact with the environment for less than $N_{\max}$ time-steps, however we define the probability measures $\mathbb{P}_{\mathcal{E}_0}$ and $\mathbb{P}_{\mathcal{E}_1}$ according to exactly $N_{\max}$ observations of the environment. If the agent stops playing at time $\tau < N_{\max}$, we simply assume the environment continues playing random actions until the end of the game.

Denote $Z = \mathbb{I}\{\tau \geq N_{\max}\}$ as the indicator of whether the agent plays for sufficiently many time steps. The condition in the lemma reads $\mathbb{P}_{\mathcal{E}_1}[Z] \geq \frac{1}{2}$. Since in environment $\mathcal{E}_0$, the agent does not receive any information about $A^*$, it holds $\sum_{t=1}^{N_{\max}} \mathbb{P}_{\mathcal{E}_0}(A_t = A^*) = N_{\max}/(K-1)$. Hence by the divergence decomposition rule of the KL divergence, we have

$$D_{KL}(\mathbb{P}_{\mathcal{E}_0} || \mathbb{P}_{\mathcal{E}_1}) = \sum_{t=1}^{N_{\max}} \mathbb{P}_{\mathcal{E}_0}(A_t = A^*) \, \mathrm{kl}\left(\frac{1}{2}, \frac{1}{2} - \Delta\right) \leq \frac{3\Delta^2 \, N_{\max}}{K-1} \leq \frac{1}{32},$$

where $\mathrm{kl}(p, q)$ denotes the KL-divergence between two Bernoulli distributions with parameters $p$ and $q$ respectively and the last inequality follows as

$$\mathrm{kl}\left(\frac{1}{2}, \frac{1}{2} - \Delta\right) = -\frac{1}{2}\ln(1 - 4\Delta^2) \le 3\Delta^2.$$

By chaining Pinsker's inequality, we have

$$\begin{aligned}
\mathbb{P}_{\mathcal{E}_1}(A_t = A^* \wedge Z) &\le \mathbb{P}_{\mathcal{E}_0}(A_t = A^* \wedge Z) + \sqrt{2D_{KL}(\mathbb{P}_{\mathcal{E}_0}||\mathbb{P}_{\mathcal{E}_1})} \\
&= \frac{\mathbb{P}_{\mathcal{E}_0}(Z)}{K - 1} + \sqrt{2D_{KL}(\mathbb{P}_{\mathcal{E}_0}||\mathbb{P}_{\mathcal{E}_1})} \\
&\le \frac{\mathbb{P}_{\mathcal{E}_1}(Z)}{K - 1} + \left(1 + \frac{1}{K - 1}\right)\sqrt{2D_{KL}(\mathbb{P}_{\mathcal{E}_0}||\mathbb{P}_{\mathcal{E}_1})}.
\end{aligned}$$

Hence

$$\mathbb{P}_{\mathcal{E}_1}(A_t = A^* \,|\, Z) \le \frac{1}{K - 1} + \frac{(1 + 1/(K - 1))\sqrt{2D_{KL}(\mathbb{P}_{\mathcal{E}_0}||\mathbb{P}_{\mathcal{E}_1})}}{\mathbb{P}_{\mathcal{E}_1}(Z)} \le \frac{1}{K - 1} + 3\sqrt{\frac{1}{16}} = \frac{3}{4}.$$

Finally, using the regret definition and combining everything

$$\mathrm{Reg} = \mathbb{E}_{\mathcal{E}_1}\left[\sum_{t=1}^{N_{\max}} \mathbb{I}\{A_t \ne A^*\}\Delta \,|\, Z\right] = \sum_{t=1}^{N_{\max}} (1 - \mathbb{P}(A_t = A^* \,|\, Z))\Delta \ge \frac{1}{4}\Delta\,N_{\max}.$$

$\square$

*Proof of Theorem 3.* Let $\mathcal{T} = x_T - 1$ be the number of switches the agent triggers from the adversary in the game. We set $\Delta = \min\left\{\frac{S\sqrt{K-1}}{3072\mathfrak{C}\sqrt{T}}, \frac{1}{8\sqrt{3}}\right\}$, which guarantees

$$N_{\max} = \left\lceil \max\left\{\frac{49152\mathfrak{C}^2 T}{S^2}, K - 1\right\} \right\rceil \le \frac{T}{2S}.$$

The regret is bounded by

$$\mathrm{Reg}_{SW}(T, S) \ge \mathbb{P}(\mathcal{T} \ne S)\frac{\Delta}{8}(T - S\,N_{\max}) = (1 - \mathbb{P}(\mathcal{T} = S))\frac{\Delta T}{16},$$

since the agent cannot have played the optimal action more than $S\,N_{\max}$ times without triggering $S$ switches. If the probability of triggering the $S$'s switch is below $\frac{1}{2}$, we are done. Otherwise by Lemma 1, the regret against a non-switching baseline on arm $K$ is bounded by

$$\mathrm{Reg}_{SW}(T, 1) \ge \mathbb{P}(\mathcal{T} = S)\frac{\Delta}{8}\,N_{\max}\,S.$$

By assumption, we have $\mathrm{Reg}_{SW}(T, 1) \le \mathfrak{C}\sqrt{(K - 1)T}$, hence $\mathbb{P}(\mathcal{T} = S) \le \frac{8\mathfrak{C}\sqrt{(K-1)T}}{\Delta\,N_{\max}\,S}$. Plugging this into the bound above, yields

$$\mathrm{Reg}_{SW}(T, S) \ge \left(1 - \frac{8\mathfrak{C}\sqrt{(K - 1)T}}{\Delta\,N_{\max}\,S}\right)\frac{\Delta T}{16} \ge \frac{\Delta T}{32} = \Omega\left(\min\left\{\frac{S}{\mathfrak{C}}\sqrt{KT}, T\right\}\right).$$

$\square$

## B.2 Stochastic lower bound for proper algorithms and proof of Theorem 4

To proof of our key lemma, Lemma 2 we first begin by showing that we can restrict our attention only to the outcome space of $(z_t)_{t=1}^{N}$, where $z_{t,i} = \ell_{t,\pi_i(x_t)}$. This done in the following lemma.

**Lemma 4.** *For any $\mathcal{E}_i, i \in [k] \cup \{0\}$ there exist a bijection from $(z_t = (\ell_{t,\pi_i(x_t)})_{i=1}^{k}, x_{t,1})$ to $(\ell_t, x_t)$. $z_t$ is a collection of independent Bernoulli random variables. The means satisfy*

$$\mathbb{E}_{\mathcal{E}_i}[z_{t,j}] = \begin{cases} \frac{1}{2} & \text{if } i \ne j \\ \frac{1}{2}(1 - \Delta) & \text{otherwise.} \end{cases}$$

*Finally $x_{t,1}$ is independent of $z_t$.*

*Proof.* The direction $(\ell_t, x_t) \to (z_t, x_{t,1})$ is given by the definition of $z_t$. The other direction is provided by

$$\ell_{t,i} = \begin{cases} z_{t,1} & \text{if } i = x_{t,1}, \\ 1 - z_{t,1} & \text{otherwise,} \end{cases}, \qquad x_{t,i} = \begin{cases} x_{t,1} & \text{if } z_{t,i} = z_{t,1} \\ 3 - x_{t,1} & \text{otherwise.} \end{cases}$$

To show the independence of $z_{t,i}$, note that by the data generation process we select $z_{t,i} = \ell_{t,\pi_i(x_t)}$ in environment $i$ such that it is a Bernoulli with mean $\frac{1}{2}(1 - \Delta)$ independent of $x_t$ (in environment $\mathcal{E}_0$, we sample $z_{t,1}$ with mean $\frac{1}{2}$). Now conditioned on $z_{t,i}$, the value of $z_{t,j}, j \neq i$ depends on whether $x_{t,i} = x_{t,j}$. $x_t$ is a collection of i.i.d. Bernoulli's with mean $\frac{1}{2}$, hence $z_{t,j}$ is independent of $z_{t,i}$ with mean $\frac{1}{2}$. We continue with the same argument over all $k$ and show that all components of $z_t$ are independent with the claimed means. Finally to show that $x_{t,1}$ is independent of $z_t$, observe that we have total symmetry over the arms $\{1, 2\}$ in our construction. Hence for any environment $i$, we have that

$$\mathbb{P}_{\mathcal{E}_i}[x_{t,1} = 1 | z_t] = \mathbb{P}_{\mathcal{E}_i}[x_{t,1} = 2 | z_t] = \frac{1}{2} = \mathbb{P}_{\mathcal{E}_i}[x_{t,1} = 1] = \mathbb{P}_{\mathcal{E}_i}[x_{t,1} = 2].$$

$\square$

We now recall the Berry-Essen inequality which we will use in the proof of Lemma 2.

**Theorem 7** (Berry-Essen inequality [Carl-Gustav, 1942, Tyurin, 2009])**.** *Let $Y_1, \ldots, Y_n$ be independent mean zero random variables second moment $\mathbb{E}[Y_i^2] = \sigma_i^2$ and absolute third moment $\mathbb{E}[|Y_i|^3] = \rho_i$. If $F_n$ denotes the CDF of $\frac{\sum_{i=1}^n Y_i}{\sqrt{\sum_{i=1}^n \sigma_i^2}}$ and $\Phi$ denotes the CDF of a standard Gaussian variable then it holds that*

$$\sup_{\alpha \in \mathbb{R}} |F_n(\alpha) - \Phi(\alpha)| \leq C \frac{\sum_{i=1}^n \rho_i}{(\sum_{i=1}^n \sigma_i^2)^{3/2}},$$

*where $C$ is some absolute constant upper bounded by 1.*

We will also need the following result relating the third central moment of a non-negative random variable to the third moment.

**Claim 1.** *Let $X \geq 0$ be a non-negative r.v. with mean $\mu > 0$. Then $\mathbb{E}[|X - \mu|^3] \leq 2\mathbb{E}[X^3]$.*

*Proof.* From triangle inequality and Jensen's inequality we have

$$\mathbb{E}[|X - \mu|^3] \leq \mathbb{E}[\max\{|X|^3, |\mu|^3\}] \leq \mathbb{E}[X^3] + \mathbb{E}[X]^3 \leq 2\mathbb{E}[X^3],$$

because $X$ and $\mu$ are non-negative. $\square$

Finally, we need the following useful TV distance inequalities.

**Lemma 5.** *For any random variables $X, Y$ (not necessarily independent) over measures $\mathbb{P}, \mathbb{Q}$ it holds that*

$$\|\mathbb{P}_{X,Y} - \mathbb{Q}_{X,Y}\|_{TV} \leq \mathbb{E}_{X \sim \mathbb{P}_X} \left[ \|\mathbb{P}_{Y|X} - \mathbb{Q}_{Y|X}\|_{TV} \right] + \|\mathbb{P}_X - \mathbb{Q}_X\|_{TV}.$$

*Proof.* Denote $s_{x,y} = \text{sign}(\mathbb{P}(X = x, Y = y) - \mathbb{Q}(X = x, Y = y))$. We have

$$\|\mathbb{P}_{X,Y} - \mathbb{Q}_{X,Y}\|_{TV} = \sum_{x,y} s_{x,y}(p(x) \cdot p(y|x) - q(x) \cdot q(y|x))$$

$$= \sum_x p(x) \sum_y s_{x,y}(p(y|x) - q(y|x)) + \sum_{x,y} s_{x,y}(p(x) - q(x))q(y|x).$$

Bounding all terms by the abs completes the proof. $\square$

*Proof of Lemma 2.* Let $\mathbb{P} = \mathbb{P}_{\mathcal{E}_0}$ be the measure induced by the algorithm under environment $\mathcal{E}_0$ and $\mathbb{Q} = \frac{1}{k} \sum_{i=1}^k \mathbb{P}_{\mathcal{E}_i}$ the mixture of measures induced by the remaining $k$ environments. We have for any event $E$

$$\min_{i \in [k]} \mathbb{P}_{\mathcal{E}_i}(E) - \mathbb{P}_{\mathcal{E}_0}(E) \leq \mathbb{Q}(E) - \mathbb{P}(E) \leq \frac{1}{2} \|\mathbb{Q} - \mathbb{P}\|_{TV}.$$

Since the random variables at time $t$ are independent of any previous time step, we can assume that the environment samples $N$ i.i.d. full-information samples $(Z, X_1) = ((z_s)_{s=1}^N, (x_{1,s})_{s=1}^N)$ and $T$ i.i.d. contexts $Y = (x_s)_{s=1}^T$ ahead of time. At any time $t$, if the agent chooses to play a revealing action, he observes the next tuple in $(Z, X_1)$, while if the agent does not play a revealing action, he observes the next element in $Y$. This construction crucially relies on the agent deciding whether to play a revealing action at time $t$ independently of $x_t$. Additional information strictly increases the total variation, so we can assume the agent always observes the full collection of random variables at the end

$$\|\mathbb{Q} - \mathbb{P}\|_{TV} \le \|\mathbb{Q}_{Z,X_1,Y} - \mathbb{P}_{Z,X_1,Y}\|_{TV} .$$

Applying Lemma 5 and observing that $\mathbb{Q}_{X_1,Y} = \mathbb{P}_{X_1,Y}$, since the contexts are sampled from the same distribution in any environment, we have

$$\|\mathbb{Q}_{Z,X_1,Y} - \mathbb{P}_{Z,X_1,Y}\|_{TV} \le \mathbb{E}_{X_1,Y\sim\mathbb{Q}}\left[\|\mathbb{Q}_{Z|X_1,Y} - \mathbb{P}_{Z|X_1,Y}\|_{TV}\right] + \|\mathbb{Q}_{X_1,Y} - \mathbb{P}_{X_1,Y}\|_{TV}$$
$$= \|\mathbb{Q}_Z - \mathbb{P}_Z\|_{TV} ,$$

where the last step follows from independence of $Z$ and $X_1, Y$. Thus we can restrict the problem to bounding the TV over $Z$.

Let $\Omega = \{0,1\}^{k\times N}$ be the outcome space of $Z$, we have

$$\frac{1}{2}\|\mathbb{Q}_Z - \mathbb{P}_Z\|_{TV} = \sum_{Z\in\Omega} \mathbb{I}\{\mathbb{Q}(Z) > \mathbb{P}(Z)\}(\mathbb{Q}(Z) - \mathbb{P}(Z))$$

For a fixed outcome $Z \in \Omega$, denote $n_i = \sum_{t=1}^N z_{i,t}$, the sum of losses of policy $\pi_i$. We have

$$\mathbb{P}(Z) = \mathbb{P}_{\mathcal{E}_0}(Z) = \left(\frac{1}{2}\right)^{Nk} \text{ and } \mathbb{P}_{\mathcal{E}_i}(Z) = \left(\frac{1}{2}\right)^{Nk}(1-\Delta)^{n_i}(1+\Delta)^{N-n_i} , \text{ hence}$$

$$\mathbb{Q}(Z) = \sum_{i=1}^k \frac{1}{k}\mathbb{P}_{\mathcal{E}_i}(Z) > \mathbb{P}(Z) \Leftrightarrow \sum_{i=1}^k \frac{1}{k}\left(\frac{1-\Delta}{1+\Delta}\right)^{n_i} > (1+\Delta)^{-N} .$$

Denote $\kappa = \ln\left(\frac{1-\Delta}{1+\Delta}\right)$. Due to symmetry, for any $i \in [k]$:

$$\mathbb{P}_{\mathcal{E}_i}\left(\sum_{h=1}^k \exp(n_h\kappa) > k(1+\Delta)^{-N}\right) = \mathbb{Q}\left(\sum_{h=1}^k \exp(n_h\kappa) > k(1+\Delta)^{-N}\right) ,$$

hence

$$\frac{1}{2}\|\mathbb{Q}_Z - \mathbb{P}_Z\|_{TV} \le \mathbb{P}_{\mathcal{E}_1}\left(\sum_{h=1}^k \exp(n_h\kappa) > k(1+\Delta)^{-N}\right) - \mathbb{P}_{\mathcal{E}_0}\left(\sum_{h=1}^k \exp(n_h\kappa) > k(1+\Delta)^{-N}\right)$$
$$= \mathbb{P}_{\mathcal{E}_0}\left(\sum_{h=1}^k \exp(n_h\kappa) \le k(1+\Delta)^{-N}\right) - \mathbb{P}_{\mathcal{E}_1}\left(\sum_{h=1}^k \exp(n_h\kappa) \le k(1+\Delta)^{-N}\right) .$$

Next we use the formula for the MGF of a Binomial r.v. $B(N, p)$ given by

$$\mathbb{E}[\exp(t\kappa k)] = (p\exp(t\kappa) + (1-p))^N,$$

to compute the expectation, variance and third moment of the r.v. $\exp(n_i\kappa)$, where $n_i$ either follows $\mathcal{E}_0$ or $\mathcal{E}_1$. Finally we will use the Berry-Essen inequality. We first compute for $m \in \mathbb{N}$

$$\mathbb{E}_{\mathcal{E}_j}[\exp(n_i m\kappa)] = \begin{cases} \left(\frac{(1+\Delta)^m + (1-\Delta)^m}{2(1+\Delta)^m}\right)^N & \text{if } i \ne j \\ \left(\frac{(1+\Delta)^{m+1} + (1-\Delta)^{m+1}}{2(1+\Delta)^m}\right)^N & \text{otherwise.} \end{cases}$$

By using $\Delta^2 N \ge \frac{\ln(k)}{40} \ge \frac{1}{2}$

$$\mathbb{E}_{\mathcal{E}_0}[\exp(n_i\kappa)] = \frac{1}{(1+\Delta)^N}$$

$$\text{Var}_{\mathcal{E}_0}[\exp(n_i\kappa)] = \frac{(1+\Delta^2)^N - 1}{(1+\Delta)^{2N}} \ge \frac{\Delta^2 N}{(1+\Delta)^{2N}} \ge \frac{1}{2(1+\Delta)^{2N}}$$

$$\mathbb{E}_{\mathcal{E}_0}[\exp(3n_i\kappa)] = \left(\frac{1+3\Delta^2}{(1+\Delta)^3}\right)^N \le \frac{\exp(3\Delta^2 N)}{(1+\Delta)^{3N}} ,$$

where we have used the facts that $(1 + x)^a \leq \exp(ax)$ and $(1 + x)^a \geq 1 + ax$, for $a \geq 1$. The Berry-Essen inequality together with Claim 1 now imply that

$$\mathbb{P}_{\mathcal{E}_0}\left[\sum_{i=1}^{k} \exp(\kappa n_i) \leq k(1+\Delta)^{-N}\right] \leq \frac{1}{2} + \frac{8 \exp(3N\Delta^2)}{\sqrt{k}}.$$

Next we compute the conditional expectation, variance and third moment for $\mathcal{E}_1$.

$$\mathbb{E}_{\mathcal{E}_1}[\exp(n_1 \kappa)] = \left(\frac{1+\Delta^2}{1+\Delta}\right)^N$$

$$\text{Var}_{\mathcal{E}_1}[\exp(n_1 \kappa)] = \left(\frac{1+3\Delta^2}{(1+\Delta)^2}\right)^N - \left(\frac{1+\Delta^2}{1+\Delta}\right)^{2N} \geq 0$$

$$\mathbb{E}_{\mathcal{E}_1}[\exp(3n_1 \kappa)] = \left(\frac{1+4\Delta^2+\Delta^4}{(1+\Delta)^3}\right)^N \leq \frac{\exp(5\Delta^2 N)}{(1+\Delta)^{3N}}$$

Let $Y_j = \exp(\kappa n_j) - \mathbb{E}_{\mathcal{E}_1}[\exp(\kappa n_j)]$, $\gamma = \frac{(1+\Delta^2)^N - 1}{(1+\Delta)^N}$. Then we have

$$-\mathbb{P}_{\mathcal{E}_1}\left[\sum_{j=1}^{k} \exp(\kappa n_j) \leq k(1+\Delta)^{-N}\right] = -\mathbb{P}_{\mathcal{E}_1}\left[\frac{\sum_{j=1}^{k} Y_j}{\sqrt{\sum_{j=1}^{k} \text{Var}_{\mathcal{E}_1}(Y_j)}} \leq -\frac{\gamma}{\sqrt{\sum_{j=1}^{k} \text{Var}_{\mathcal{E}_1}(Y_j)}}\right]$$

$$\leq -\Phi\left(-\frac{\gamma}{\sqrt{\sum_{j=1}^{k} \text{Var}_{\mathcal{E}_1}(Y_j)}}\right) + \frac{8\exp(5N\Delta^2)}{\sqrt{k-1}},$$

where in the inequality we have used the Berry-Essen inequality. To bound the remaining term, we have

$$\frac{\gamma}{\sqrt{\sum_{j=1}^{k} \text{Var}_{\mathcal{E}_1}(Y_j)}} \leq \sqrt{\frac{(1+\Delta^2)^N - 1}{k-1}}.$$

Let $X$ be a standard Normal r.v., then to bound $-\Phi\left(-\frac{\gamma}{\sqrt{\sum_{j=1}^{k} \text{Var}_{\mathcal{E}_1}(Y_j)}}\right)$ we have

$$\Phi\left(-\frac{\gamma}{\sqrt{\sum_{j=1}^{k} \text{Var}_{\mathcal{E}_1}(Y_j)}}\right) = \mathbb{P}\left(X \geq \frac{\gamma}{\sqrt{\sum_{j=1}^{k} \text{Var}_{\mathcal{E}_1}(Y_j)}}\right) \geq \frac{1}{2} - \sqrt{\frac{(1+\Delta^2)^N - 1}{k-1}},$$

where in the inequality we decomposed the tail probability into the integral from $0$ to $\infty$ minus the integral from $0$ to $\sqrt{\frac{\gamma}{k}}$. Combining the above bounds we have that the TV is bounded by

$$\frac{1}{2}\|\mathbb{Q}_Z - \mathbb{P}_Z\|_{TV} \leq 8\exp(3\Delta^2 N - \ln(k)/2) + 8\exp(5\Delta^2 N - \ln(k-1)/2) + \exp(\Delta^2 N - \ln(k-1)/2).$$

Finally, we use $\Delta^2 N \leq \lfloor\frac{\ln(k-1)}{20\Delta^2}\rfloor \leq \frac{\ln(k-1)}{20}$ and $k > e^{20}$ to obtain

$$\frac{1}{2}\|\mathbb{Q}_Z - \mathbb{P}_Z\|_{TV} \leq 17\exp(-\ln(k-1)/4) = \frac{17}{\sqrt[4]{k-1}} \leq \frac{1}{4}.$$

$\square$

With Lemma 2 we are ready to give the proof of Theorem 4.

*Proof of Theorem 4.* Let $c_1 = \frac{1}{160}, c_2 = \frac{1}{10}c_1^{-2}$. We assume w.l.o.g. that $c_2\mathfrak{C}^2 \leq \ln(k) \leq \frac{1}{2}T$. If $\ln(k) = \mathcal{O}(\mathfrak{C}^2)$, then the regret is lower bounded by the regret of $\Pi_1$, if $\ln(k) = \Omega(T)$, the optimal regret is linear in $T$ anyway. Pick $\Delta = \min\{\frac{c_1 \ln(k)}{\mathfrak{C}\sqrt{T}}, \frac{1}{4}\}$. This choice implies $N = \lfloor\frac{\ln(k)}{20\Delta^2}\rfloor \leq \frac{T}{2}$.

Denote $\mathcal{N}$ the random number of plays in action $\{1,2\}$, i.e. the number of observations of the full information game. The regret in environment $\mathcal{E}_0$ is given by

$$\mathrm{Reg}_{\mathcal{E}_0}(T,\Pi_1) = \mathbb{E}_{\mathcal{E}_0}[\mathcal{N}]\frac{\Delta}{4} \le \mathfrak{C}\sqrt{T} \le \frac{c_1 \ln(k)}{\Delta}\,.$$

Hence

$$\mathbb{E}_{\mathcal{E}_0}[\mathcal{N}] \le 4\frac{c_1 \ln(k)}{\Delta^2} \le 80 c_1 N\,.$$

Given any algorithm $\mathcal{A}$, we create a modified algorithm $\underline{\mathcal{A}}$ that uses a stopping time to commit to action 3 after it played $N$ times action $\{1,2\}$. By Markov inequality, the probability of $\mathcal{A}$ hitting the stopping time on $\mathcal{E}_0$ is bounded by $80 c_1 \le \frac{1}{2}$. Denote that event by $E$. By choice of our constants, the conditions for Lemma 2 are met, which implies

$$\mathbb{P}_{\mathcal{E}_{i^*}}(E) := \min_{i\in[k]} \mathbb{P}_{\mathcal{E}_i}(E) \le \frac{1}{2} + \frac{1}{4} = \frac{3}{4}\,.$$

Hence

$$\mathrm{Reg}_{\mathcal{E}_{i^*}}(T,\Pi_2,\mathcal{A}) \ge \frac{1}{4}\mathrm{Reg}_{\mathcal{E}_{i^*}}(T,\Pi_2,\underline{\mathcal{A}}) \ge \frac{1}{4}(T-N)\frac{\Delta}{4} \ge \frac{\Delta T}{32} = \Omega\left(\min\left\{\frac{\ln(k)\sqrt{T}}{\mathfrak{C}}, T\right\}\right)\,.$$

$\square$

## B.3 Stochastic lower bound for improper algorithms and proof of Theorem 5

We first show the counterpart to Lemma 2 for improper algorithms. The proof uses Lemma 2 together with a sort of a union bound over all possible realizations of time steps at which an improper algorithm chooses to observe the full information game.

*Proof of Lemma 3.* As in the proof of Lemma 2, we denote $\mathbb{P} = \mathbb{P}_{\mathcal{E}_0}$ the measure induced by running the algorithm in environment $\mathcal{E}_0$ and $\mathbb{Q} = \frac{1}{k}\sum_{i=1}^{k}\mathbb{P}_{\mathcal{E}_i}$ the mixture over measures induced by environments $i \in [k]$. $\mathbb{P}$ and $\mathbb{Q}$ are measures over all possible outcomes of the random variables $Z = (z_t)_{t=1}^T, X = (x_t)_{t=1}^T, \mathcal{T} = (\tau_i)_{i=1}^N$. Let for any subset $M \subset [T]$ denote $Z(M) = (z_t)_{t\in M}$, then the observations of the algorithms are $X, \mathcal{T}, Z(\mathcal{T})$. We have

$\|\mathbb{Q}_{X,\mathcal{T},Z(\mathcal{T})} - \mathbb{P}_{X,\mathcal{T},Z(\mathcal{T})}\|_{TV}$

$$= \sum_{M\in\mathcal{N}, z\in\{0,1\}^{Nk}, x\in\{1,2\}^{Tk}} |\mathbb{Q}(\mathcal{T}=M, Z(M)=z, X=x) - \mathbb{P}(\mathcal{T}=M, Z(M)=z, X=x)|$$

$$= \sum_{M\in\mathcal{N}, z\in\{0,1\}^{Nk}, x\in\{1,2\}^{Tk}} \Big(|\mathbb{Q}(\mathcal{T}=M|Z(M)=z, X=x)\cdot\mathbb{Q}(Z(M)=z, X=x)$$

$$- \mathbb{P}(\mathcal{T}=M|Z(M)=z, X=x)\cdot\mathbb{P}(Z(M)=z, X=x)|\Big)$$

$$= \sum_{M\in\mathcal{N}, z\in\{0,1\}^{Nk}, x\in\{1,2\}^{Tk}} \mathbb{Q}(\mathcal{T}=M|Z(M)=z, X=x)|\mathbb{Q}(Z(M)=z, X=x) - \mathbb{P}(Z(M)=z, X=x)|$$

$$\le \sum_{M\in\mathcal{N}} \|\mathbb{Q}_{Z(M),X} - \mathbb{P}_{Z(M),X}\|_{TV}\,,$$

where we use in the third equality that both measures are induced by the same algorithm and hence the conditional probabilities satisfy

$$\mathbb{Q}(\mathcal{T}=M|Z(M)=z, X=x) = \mathbb{P}(\mathcal{T}=M|Z(M)=z, X=x)\,.$$

We use Lemma 2 to finish the proof. $\square$

*Proof of Theorem 5.* Let $\Delta = \frac{1}{4}$ and $N = \lceil 32\mathfrak{C}\sqrt{T}\rceil$. Following the proof of Theorem 4 we can reduce the problem to a player which plays an algorithm that commits to action 0 after $N$ observations of the full game. Using Lemma 3 we can bound

$$\frac{1}{2}\|\mathbb{Q}_{\mathcal{T},Z(\mathcal{T}),X} - \mathbb{P}_{\mathcal{T},Z(\mathcal{T}),X}\|_{TV} \le 17\exp\left(N\log(T) - \frac{\log(k-1)}{4}\right)\,.$$

It is sufficient to set $k = 1 + 68\exp(4(1 + 32\mathfrak{C}\log(T)\sqrt{T}))$ to ensure that then $\frac{1}{2}\|\mathbb{Q}_{\mathcal{T},Z(\mathcal{T}),X} - \mathbb{P}_{\mathcal{T},Z(\mathcal{T}),X}\|_{TV} \le 1/4$. We can then proceed as in the proof of Theorem 4. $\square$

## B.4   Impossibility of second order bounds

*Proof of Theorem 6.* Given an algorithm $\mathcal{A}$ satisfying Eq. (4), we construct the following wrapper. Initialize $\mathcal{A}$ with $\Pi_2$ augmented with a total of $k$ copies of $\pi_0$. At any time $t$ receive $p_t \in \Delta(\Pi_2)$ (collapsing the copies of $\pi_0$). With probability $\gamma = 3T^{-1/2+\alpha/2}$, play one of the 3 actions uniformly at random (we do this by following $\pi_0$ with probability $\gamma/3$ and following $\pi_1$ with probability $2\gamma/3$, ensuring a proper algorithm), otherwise sample $\pi_t \sim p_t$ and play $A_t = \pi_t(x_t)$. Observe $\ell_{t,A_t}$ and construct loss estimator

$$\hat{\ell}_t = \frac{\sum_{\pi \in \Pi_2 : \pi(x_t)=A_t} \ell_{t,A_t} \mathbf{e}_\pi}{\gamma/3 + (1-\gamma) \sum_{\pi \in \Pi_2 : \pi(x_t)=A_t} p_{t,\pi}} .$$

Finally feed $\hat{\ell}_t \cdot \gamma/3$ to $\mathcal{A}$. By construction, the wrapper is *proper* and the loss range for the base algorithm is bounded in $[0,1]$. The variance terms encountered by $\mathcal{A}$ are

$$\mathbb{V}_{\pi \sim p_t}[\frac{\hat{\ell}_{t,\pi}\gamma}{3}] \leq \frac{\gamma^2}{9} \sum_{i=0}^{k} p_{t,\pi_i} \left( \frac{\mathbb{I}\{\pi_i(x_t)=A_t\}\ell_{t,A_t}}{\gamma/3 + (1-\gamma)\sum_{j:\pi_i(x_t)=\pi_j(x_t)} p_{t,\pi_j}} \right)^2$$

$$\leq \frac{\gamma^2}{9(1-\gamma)(\gamma/3 + (1-\gamma)\sum_{j:\pi_j(x_t)=A_t} p_{t,\pi_j})} ,$$

and in expectation over the choice of $A_t$, we have

$$\mathbb{E}[\mathbb{V}_{\pi \sim p_i}[\frac{\hat{\ell}_{t,\pi}\gamma}{3}]] \leq \frac{\gamma^2}{9(1-\gamma)} K \leq \frac{2}{3}\gamma^2 .$$

The regret in environment $\mathcal{E}_0$ satisfies due to unbiasedness for selecting the top $1/2$ of all policies (which are the copies of $\pi_0$)

$$\text{Reg}(T, \Pi_1) = \mathbb{E}\left[ \sum_{t=1}^{T} \ell_{t,A_t} - \hat{\ell}_{t,\pi_0(x_t)} \right] \leq \gamma T + \frac{3(1-\gamma)}{\gamma} \mathbb{E}\left[ \sum_{t=1}^{T} \langle p_t - u_{\frac{1}{2}}, \gamma\hat{\ell}_t/3 \rangle \right]$$

$$= \gamma T + \frac{3(1-\gamma)}{\gamma} \mathbb{E}\left[ \tilde{\mathcal{O}}\left( \sqrt{\sum_{t=1}^{T} \left( \mathbb{V}_{\pi \sim p_t}[\gamma\hat{\ell}_{t,\pi}/3] + T^\alpha \right) \ln(2)} \right) \right] = \tilde{\mathcal{O}}\left( T^{(1+\alpha)/2} \right) ,$$

where the last step is by Jensen's inequality. Analogously, for any environment $\mathcal{E}_i, i \in [k]$, we have for $\epsilon = 1/(2k)$

$$\text{Reg}(T, \Pi_2) = \tilde{\mathcal{O}}\left( T^{(1+\alpha)/2} + \sqrt{T \ln(|\Pi_2|)} \right) .$$

Following the same argument as in the proof of Corollary 1, an upper bound of $\tilde{\mathcal{O}}(T^{(1+\alpha)/2})$ implies $\mathfrak{C} = \tilde{\Theta}(T^{\alpha/2})$, hence by the proof of Theorem 4, we have

$$\text{Reg}(T, \Pi_2) = \tilde{\Omega}\left( \frac{\ln(|\Pi_2|)}{\mathfrak{C}} \sqrt{T} \right) = \tilde{\Omega}\left( \ln(|\Pi_2|) T^{(1-\alpha)/2} \right) .$$

This leads to a contradiction for $\ln(|\Pi_2|) = \Omega(T^{(1+\alpha)/2})$. $\qquad\square$