# OpenReview forum: "The Pareto Frontier of model selection for general Contextual Bandits"
_NeurIPS.cc/2021/Conference — NeurIPS 2021 Poster_

### Official Review · Reviewer_d1Ft · 2021-07-16

**Rating:** 8
**Confidence:** 4

**Summary:**

This paper studies the model selection problem in contextual bandits. Lower bounds for the model selection problem, the switching rewards problem, and the second-order bound in the full-information game are provided, which resolve several open problems.

**Limitations And Societal Impact:**

As discussed in the checklist section.

**Main Review:**

Lower bounds established in this paper resolve several open problems, i.e., impossible results for model selection under bandit feedback, impossible results for adapting to the number of switches in an adversarial setting, and the second-order bound for an \epsilon portion of experts in the full information setting. I believe these results are important to the community. Although I didn't carefully check the proofs, the theoretical results stated in the paper make sense to me.

One minor drawback is that the lower bound for the stochastic setting is with respect to proper algorithms (Definition 1). However, the statements made by the authors (line 276-290) is compelling to me.

====after rebuttal====

I will keep my score at 8. However, I would like to see the clarification questions raised by Reviewer tzvm to be addressed in the next version of the paper.

**Time Spent Reviewing:**

4

---

> ### Author Response · Authors · 2021-08-10
> **Reply**
>
> We would like to thank the reviewer for their evaluation of our work.

---

### Official Review · Reviewer_KX4F · 2021-07-16

**Rating:** 5
**Confidence:** 3

**Summary:**

This paper studies the model selection for contextual bandits. It approaches two open problems. One is a COLT2020 open problem of Foster et al, which askes if there exists a constant such that a regret bound (1) holds for all m \in [0,M] simultaneously for a nested sequence of a finite set of policies m = 1, ..., M . The other one is a COLT2016 open problem of Freund asking if there is an algorithm for the online full feedback problem with adversarial losses that achieves \sqrt{V_T logK} regret, where V_T is the cumulative variance across experts. The main claim of the paper is negative answers on both open problems.

**Limitations And Societal Impact:**

There is no potential negative societal impact.

**Main Review:**

Before discussing the paper's results and methods, my main caution is that the paper is not written so that the ideas can be directly understood from the text. First of all, it feels that a solid block of definitions is missed before page 5. For example, on page 5, the Algorithm 1 and variables in the text appear suddenly and are not defined later in the paper, and there is no reference to other papers where it could be found.


Regarding the stated results, looking at the proof of Theorem 5, I do not get where the loss estimate comes from, especially since no loss estimate is required in a full information setting.

I put my Confidence score 3 as I don't get many claims in the paper because there is no definition for many variables.

**Time Spent Reviewing:**

5

---

> ### Author Response · Authors · 2021-08-10
> **Missed definitions and Theorem 5**
>
> > First of all, it feels that a solid block of definitions is missed before page 5. For example, on page 5, the Algorithm 1 and variables in the text appear suddenly and are not defined later in the paper, and there is no reference to other papers where it could be found.
>
> A detailed version of the corralling algorithm can be found in the cited work of Foster et al. 2020 (Appendix D). The algorithm presented in our work is essentially the same with the exception of replacing R by the vector scaled quantity defined on line 167.
>
> We now clarify the notation and explain a typo which might have confused the reviewer. First we note that $p_t$ should be $q_t$ in the description of the algorithm and this is the distribution which the corralling algorithm keeps over base algorithms. $\alpha$ is a parameter which depends on the “stability” of the base algorithms (see lines 178-179 and the work of Agarwal et al. 2017). R is a vector of regret upper bounds for the base algorithms (see line 167). $\beta$ is a vector of parameters with $m$-th coordinate depending on the complexity of the decision space of the $m$-th base algorithm (see line 186-187). $Top$ and $(Base)_\{m=1\}^M$ are respectively the corralling and base algorithms. A general description of our algorithm can be found in lines 170-184. The sampling distribution of the corralling algorithm is generated by the FTRL update as $q_\{t+1\} = \text{arg}\min_\{q\in\Delta^\{M-1\}\} \langle q,L_t - B_t \rangle +\frac{F(q)}{\eta}$ where $B_\{t,m\}$ is the LHS of Equation 2, $F$ is the potential induced by the ½-Tsallis entropy and $\eta$ is an appropriately set step size (line 466 in the Appendix). If space permits we are happy to have an extended description of the above algorithm similar to what is in Foster et al. 2020 (Appendix D) and we can include a full algorithm with pseudo-code in the appendix of the current work if the reviewer thinks this would improve the presentation of the paper. The reason we did not go into a detailed description of the corralling algorithm in the main text is because we think the main contributions are in terms of our lower bounds and prefer discussing those results at length.
>
> > Regarding the stated results, looking at the proof of Theorem 5, I do not get where the loss estimate comes from, especially since no loss estimate is required in a full information setting.
>
> The proof of Theorem 5 works by showing that if there exists an algorithm $\mathcal{A}$ which solves the COLT2016 problem then we can use this algorithm to construct an algorithm for the model selection problem with a regret bound which contradicts Theorem 4. While $\mathcal{A}$ is an algorithm for a full information game we use it to solve the contextual bandit problem, similarly to how one can solve the bandit problem using Hedge. That is, one feeds unbiased estimators of the losses to the Hedge algorithm and we end up with a version of Exp3. The reduction works as follows. Given the model selection problem in Theorem 4 we initialize a copy of A with $2k$ actions. Each action either corresponds to one of the $k$ policies in $\Pi_2$ or to one of $k$ copies of policy $\pi_0$. At every round of the game A has a distribution over the extended policy class and samples a policy to be played. After which we observe the context $x_t$ and select the action according to the sampled policy. This allows us to construct an estimator of the loss of all policies (actions of $\mathcal{A}$) as done on page 17 of the appendix. We feed an appropriately rescaled version of this loss estimator to $\mathcal{A}$ so that all losses are in $[0,1]$. A then updates its internal state and produces a new distribution over policies and the game proceeds to the next round. Because $\mathcal{A}$ does not use the context $x_t$ at time $t$ to make a choice of policy it is proper. The regret bound on $\mathcal{A}$ now implies a regret bound for the model selection problem which contradicts Theorem 4.

---

> > ### Comment · Reviewer_KX4F · 2021-08-27
> > **reply**
> >
> > Thank you for your explanation, it makes total sense now. Anyway, there are still many notions in the paper that should be defined, and there are definitely problems with clarity, so I would not increase the rating much. However, I am looking forward to seeing a reworked version of this paper.

---

### Official Review · Reviewer_9QgL · 2021-07-17

**Rating:** 7
**Confidence:** 3

**Summary:**

The paper considers a model selection problem in contextual bandits, and resolve a few open questions regarding the achievable regret bounds. For finite sized policy classes, a Pareto frontier of upper bounds for model selection is provided. With an additional assumption of the algorithms being proper, matching lower bounds are provided as well. The paper also considers the case of adaptive adversarial problems, and discuss the challenges involved.

The paper is well written with clear context and literature overview. While the algorithmic contribution is incremental, the proof techniques for the lower bounds are interesting and add value to the literature.

**Limitations And Societal Impact:**

 While a motivation for the work is to resolve open problems proposed previously, it would be helpful to give an example application and illustrate the challenges and contribution.

**Main Review:**

The related literature section can be improved by including the definitions and intuition behind the symbols used in regret expressions.

Also, it would be useful to include the precise statements of the open problems in introduction, and what are the challenges involved. And why is it important to address them.

Questions:
1. Does the policy class include situations where the optimal policy can outperform a policy that plays the optimal fixed arm?
2. In case of adversarial switching, what is the the optimal oracle policy in the regret?
3. Suppose there is a known underlying structure that dictates the switching, can this help in relaxing the use of proper algorithms to reduce the technicalities for lower bound proofs?



**Time Spent Reviewing:**

3

---

> ### Author Response · Authors · 2021-08-10
> **Reply**
>
> We thank the reviewer for the suggestions for improving clarity by introducing appropriate notation used in regret bounds and having formal statements of the open problems. We can include a separate notation section after the introduction and we can formally state the open problems with discussion of significance if there is enough space in the final version of the paper.
>
> We now answer the questions:
>
> > Does the policy class include situations where the optimal policy can outperform a policy that plays the optimal fixed arm?
>
> Even in our stochastic lower bound it is not necessarily the case that a policy which plays a fixed action is the best overall. Indeed, by construction the contexts are sampled so that the optimal policy has an equal chance of playing any action at the current round. Thus it is important to identify the optimal policy rather than a fixed arm.
>
> > In case of adversarial switching, what is the the optimal oracle policy in the regret?
>
> The optimal policy for the S-switch problem is the one which switches S times and always plays the optimal arm for the current switch phase.
>
> > Suppose there is a known underlying structure that dictates the switching, can this help in relaxing the use of proper algorithms to reduce the technicalities for lower bound proofs?
>
> We are not sure if we understand the question. Properness is only required for the stochastic lower bound, which is not about switching. In fact, both in the oblivious and stochastic case, the s-switch problem is provably easier than the general model selection problem and our lower bound does not apply.
>
> > While a motivation for the work is to resolve open problems proposed previously, it would be helpful to give an example application and illustrate the challenges and contribution.
>
> Our negative results show that all prior work on model selection for contextual bandits in fact needs the additional assumptions and model selection in general is impossible. It would be interesting to investigate more general settings in which model selection is possible, e.g., contexts are correlated as this would break our construction. The main challenge we faced was in the lower bound proof for the stochastic case as standard information theoretic techniques which go through bounding the TV through KL divergence do not really yield anything informative. In fact it is very hard to directly compute the KL divergence between our mixture of distributions and $\mathbb{P}_{\mathcal{E}_0}$. To the best of our knowledge, directly bounding the TV and the use of Berry-Esseen is novel.

---

### Official Review · Reviewer_tzvm · 2021-07-24

**Rating:** 8
**Confidence:** 4

**Summary:**

The paper studies a COLT 2020 open problem on model selection for contextual bandits (Foster et al. 2020b). The paper presents strong lower bounds which are claimed to hold for all algorithms in the adaptive adversarial regime and hold for "proper" algorithms in the stochastic regime. If the lower bounds are true, then they would lead to the following consequences:
1.  The paper resolves the open problem of Foster et al. (2020b) by providing a negative answer.
2.  As a byproduct, the paper resolves a COLT 2016 open problem on second-order bounds for full-information online learning (Freund 2016).

For the model selection problem, the paper also provides a matching upper bound, thus gives a complete characterization of the pareto frontier of model selection for contextual bandits. The paper also discusses the implications to the $S$-switch non-stationary bandit problem.

**Ethical Concerns:**

This paper is theoretical in nature and I do not foresee any immediate societal impacts.

**Limitations And Societal Impact:**

I have specified my concerns above, which may make the current version of the paper fail to obtain the claimed results. That being said, if the author can convince that there is no issue in the current version, then I would be very willing to give a high score.

By the way, I spent some time thinking about the construction for the stochastic lower bound, and cannot come up with a model selection algorithm which overcomes this hard instance. While I think that the current proof techniques may be not sufficient to lead to the desired result, I feel that the author is in fact in the right direction of constructing hard instances, and it is possible that this instance indeed leads to the desired lower bound (with more advanced proof techniques).

**Main Review:**

========== After rebuttal

I have read the authors' reparametrization argument in their response, which seems like a clever way to simplify the outcome space, and seems to address my major concern. After adding this argument, the lower bound proof implicitly utilizes the component-wise independence of $x_t$ in a substantial way, which seems to be crucial. While I may check more details later, I believe that the proof makes sense (after adding the non-obvious reparametrization argument which was completely missing in the original version). I thus believe that the paper makes significant contribution to the literature and would like to change my score. While I strongly recommend acceptance based on the theoretical contribution, I hope that the authors can add the missing arguments to the main text and spend time improving the presentation, as the original submission is very difficult to understand.

========== Original review

The paper considers a challenging problem; if the claimed results are true, then the paper would definitely be a breakthrough in this field. I carefully read the paper, and spent lots of time verifying the results. I have a mixed feeling on this paper. On the one hand, the paper seems to contain several interesting and potentially important ideas. On the other hand, I have a major concern on the soundness of a main (and in my opinion the most important) result; see the "major concern" part for details. As a result, I cannot give a good score. However, if the author can convince me that there is no issue, then I would definitely be happy to give a high score.

By the way, the writing of the current paper is not good: there are many typos and the main technical ideas are somehow hard to follow (although I am familiar with the problem of model selection for contextual bandits, I had to figure out and fill in lots of details on my own). Regardless of whether this paper can be accepted, I hope that there can be a significant improvement on writing if there is a next version.

----- Major concern

In my opinion, the stochastic lower bound in Section 4.3 is the most important result of this paper. While other results including the upper bounds in Section 3 and the adaptive adversarial lower bound in Section 4.1 are also interesting, they are not as exciting as the stochastic lower bound (the upper bounds are obtained by applying and modifying existing techniques, while the adaptive adversarial lower bound is more restrictive compared with the stochastic lower bound). Therefore, I paid the most attention to the stochastic lower bound.

While requiring a "proper" algorithm is definitely not a weak condition (as it precludes many interesting algorithms like the $\epsilon$-greedy algorithm), this is not a problem for me as a lower bound for proper algorithms is already sufficient to imply a strong lower bound for Freund's COLT 2016 problem. I would thus be willing to give the paper a high score as long as Theorem 4 is correct.

My major concern is that the current version of Lemma 2 is not sufficient to support Theorem 4. In particular, since the proof of Lemma 2 implicitly assumes that the considered event is defined on the space of $Nk$ Bernoulli random variables (see line 501) which is the space of all loss vectors, the event $E$ appearing in the statement of Lemma 2 can only be an event completely determined by the realizations of loss vectors (otherwise $E$ would be non-measurable). As a result, $E$ cannot be an algorithm-dependent event in general, as an algorithm can make decisions based on not only the observed losses but also the *observed contexts* --- while the author does assume a “proper” algorithm, being proper only means that the algorithm cannot use $x_t$ to decide $\pi_t$ (by the way, there are also some issues with the current definition of being "proper"; see the "other concerns" section), but does not mean that the algorithm cannot use $x_1,\dots,x_{t-1}$ to decide $\pi_t$. As a result, the current proof of Theorem 4 seems to be problematic --- since the event $E$ defined in the proof of Theorem 4 is algorithm-dependent (which means that the event depends on the *realizations of contexts*), it is not a measurable event required by Lemma 2, so Lemma 2 does not tell anything about $E$. In other words, since the current proof of Lemma 2 crucially relies on the fact that $E$ only depends on the *realizations of loss vectors*, it seems to me that the author did not really prove a lower bound for all "proper algorithms", but proved a much weaker lower bound which only holds for all "algorithms that cannot observe contexts at all" --- this makes the actually proved lower bound not interesting as almost all reasonable contextual bandit algorithms rely on the use of context information (otherwise the problem is not "contextual" and is more similar to semi-bandits or bandits with many arms). I also spent quite some time thinking about the current main technique used to prove Lemma 2 (i.e., the Berry-Essen-based argument), and find it difficult to extend to the "normal" case where the algorithm can use the information of contexts. In particular, since Berry-Essen only holds for *independent* random variables, the current proof technique critically relies on the fact that "at time $t$, the loss observed by policy $\pi_1$ and the loss observed by policy $\pi_2$ is independent". However, if the algorithm can observe $x_t$, then this independence does not hold conditional on the observation of $x_t$ --- while being proper might suggest that the algorithm would not use $x_t$ to decide the policy used in round $t$, the algorithm can always use the information of $x_t$ and $\ell_{t,A_t}$ at round $t+1$, so the current technique based on independence might be insufficient to address the challenge.

I make two additional comments. First, if the author cannot address the issue that I state above, then the claim that Freund's COLT 2016 open problem is solved by the paper is false. Note that the algorithm constructed by the author in Appendix B.2 does use the realization of historical contexts to decide the policy used at round $t+1$ (see line 526), so this algorithm does not violate the actual lower bound that the author is able to prove in Theorem 4. Second, while I might misunderstand something, I feel the current proof of the lower bound extremely hard to follow, as the author does not really clearly specify the probability spaces when trying to prove "change of measure"-type results --- in my opinion, any rigorous proof should be clear about key definitions, and the current writing of the proof requires significant improvement.

----- Other concerns
1. Definition 1: The current definition of a "proper" algorithm is a little problematic. The author should add a restriction that $\pi_t\in\Pi_M$ to ensure that this definition is meaningful; otherwise, any algorithm can satisfy this definition --- if $\pi_t$ is arbitrary, then any algorithm can be thought as a procedure which sequentially selects $\pi_t$ independent of $x_t$.
2. Line 526: The author should try to avoid using $x_{t,i}$ as a notation for the realized loss, as $x_t$ also stands for a context.
3. Lines 503-503: $(1-\Delta)$ is a typo and should be changed to $(1+\Delta)$.
4. Line 517: Since the author is trying to bound the distance between the measures induced by ${\mathcal{E}_0}$ and ${\mathcal{E}_1}$, the current TV distance must be a typo (as it involves $\mathcal{E}_1$ and $\mathcal{E}_2$). Also, it should not be TV distance. There must be some constraints as specified in the earlier parts of the proof.
5. Line 518: there is a typo on the place where $\Delta^2$ appears.

**Time Spent Reviewing:**

14

---

> ### Author Response · Authors · 2021-08-10
> **Addressing major concern**
>
> We would like to thank the reviewer for his outstanding amount of time he spent reviewing this paper and especially that he verified the correctness of our proof conditioned on the single concern. We are confident that we have a rigorous answer to this concern, showing that our proof is indeed correct.
> We further like to thank the reviewer for noting the typos, and he is correct that we mean $\pi_t\in\Pi_M$. (In fact, what we need is that the agent must commit to whether $A_t$ is a revealing action ahead of seeing the context, which is satisfied by $\pi_t \in \Pi_M$, but also for epsilon-greedy which was mentioned by the reviewer.)
>
> To address his concern about using Berry-Esseen, we want to remind the reviewer of our comment about reparametrization and elaborate in case it was not sufficiently clear in our paper. The environmental random variables at each time-step are the context $x_t$ representing the action any policy takes at time $t$ (for simplicity, let $x_t\in \\{1,2\\}^k$, and $\pi_i(x_t) = x_{t,i}$), and the loss vector $\ell_t \in \\{0,1\\}^2$ for the loss of action $1$ and $2$ respectively (the loss of action 3 is not random).
> We can reparameterize this random variable by $x_{t,1}$ and $z_t\in\\{0,1\\}^k$, where $z_{t,i} = \ell_{t, \pi_i(x_t)}$ is the loss vector of the $k$ policies. $z_t$ together with $x_{t,1}$ is a valid reparametrization, in the sense that there exists a bijection of $(x_t,\ell_t)$ to $(z_t, x_{t,1})$. The other direction of this bijection is $x_{t,i} = x_{t,1}$ if $z_{t,i}=z_{t,1}$ and $x_{t,i}=3-x_{t,1}$ otherwise, while $\ell_{t,x_{t,1}} = z_{t,1}, \ell_{t, 3-x_{t,1}} = 1-z_{t,1}$.
>
> We noted in our paper that under this reparametrization, the random variable $z_t$ is a collection of independent Bernoulli random variables. This can be seen by the data generation process.
> We start by sampling the (if existing) biased component $z_{t,i}$ together with its context $x_{t,i}$. Now conditioned on $z_{t,i}$, the value for any $z_{t,j}$ ($j\neq i$) is a $0.5$-bernoulli random variable since $x_{t,j}$ is an i.i.d. unbiased Bernoulli and the loss is determined by whether $x_{t,j}=x_{t,i}$ holds. Hence $z_{t,j}$ is independent of $z_{t,i}$ (to see this factorize the joint distribution by first conditioning on $z_{t,i}$). Going forward we sample one component of $z_t$ after the other independent of all previous ones. This results in the vector $z_t$ being a collection of independent Bernoullis.
> This shows the validity of using Berry-Esseen for the random variable $z_t$ (in the proof of Lemma 2, we refer to the random variable $z$ by $x$, which is indeed a confusing choice of notation.)
> $x_{t,1}$ is also independent of $z_t$. Denote $\bar x_t$ as the context where all arms are switched compared to $x_t$. We have by construction for any environment $\mathbb{P}(z_t|x_t)=\mathbb{P}(z_t|\bar x_t)$. Together with $\mathbb{P}(x_t)=\mathbb{P}(\bar x_t)$, this implies $\mathbb{P}(x_t|z_t)=\mathbb{P}(\bar x_t|z_t)$. Finally to get the conditional distribution of $x_{t,1}$, we have
> $\mathbb{P}(x_{t,1}=1|z_t)=\sum_{x:x_{1}=1}\mathbb{P}(x|z_t)=\sum_{x:x_{1}=1}\mathbb{P}(\bar x|z_t)=\mathbb{P}(x_{t,1}=2|z_t)$.
> Hence $x_{t,1}$ is independent of $z_t$.
>
> > My major concern is that the current version of Lemma 2 is not sufficient to support Theorem 4. In particular, since the proof of Lemma 2 implicitly assumes that the considered event is defined on the space of $Nk$ Bernoulli random variables (see line 501) which is the space of all loss vectors (including potential outcomes), the event $E$ appearing in the statement of Lemma 2 can only be an event completely determined by the realizations of loss vectors (otherwise $E$ would be non-measurable). As a result, $E$ cannot be an algorithm-dependent event in general, as an algorithm can make decisions based on not only the observed losses but also the observed contexts
>
> The reviewer’s main concern is that we only calculate the total variation over the outcome space of $Z=(z_1,\dots z_{N})$ instead of considering the contexts. We justified this reduction to $Z$ by noting that the contexts without losses are uninformative since they are drawn independently from a distribution independent of the environment. We believe that to be a standard argument in information theory. Let us elaborate. Since the agent makes the choice of whether to observe all information, i.e. $z_t, x_{t,1}$, or only $x_t$ ahead of time $t$ and all data over different times is i.i.d., we can without loss of generality assume that the environment samples two batches of data ahead of time. $(Z, X_1) = ((z_{1},x_{1,1}),\dots (z_N,x_{N,1}) )$ and $Y = (x_{N+1},\dots X_{T+N})$. At any time, if the agent observes all information (which happens at most $N$ times), the agent receives the next element in $(Z,X_1)$, otherwise the agent receives the next unseen element in $Y$.
>
> Now it is clear that bounding the TV between environments over the outcomes of $(Z,X_1,Y)$ would be sufficient for the proof.
> We have also just seen that $(X_1,Y)$ is drawn independently of $Z$ and it is drawn from the same distribution regardless of the environment $\forall i\in[k]\cup\\{0\\}:\,\mathbb{P}_i(X_1,Y)=\mathbb{P}(X_1,Y)$. Hence we have $TV( \mathbb{P}_i,\mathbb{P}_j) = \sum_\{(Z,X_1,Y)\}|\mathbb{P}_i(Z,X_1,Y)-\mathbb{P}_j(Z,X_1,Y)|= \sum_\{(Z,X_1,Y)\}|\mathbb{P}_i(Z)\mathbb{P}_i(X_1,Y)-\mathbb{P}_j(Z)\mathbb{P}_j(X_1,Y)|=$
> $\sum_\{Z\}\sum_\{(X_1,Y)\}\mathbb{P}(X_1,Y)|\mathbb{P}_i(Z)-\mathbb{P}_j(Z)|
> = \sum_\{Z\}|\mathbb{P}_i(Z)-\mathbb{P}_j(Z)|.
> $
> In other words, the total variation we are bounding in Lemma 2 over the outcomes of $Z$ is all that matters.
>
> We will include the above discussion in the paper and fix the confusing notation.

---

> > ### Comment · Reviewer_tzvm · 2021-08-22
> > **Congratulations on your nice solution**
> >
> > Hi, I have read the reparametrization argument in your response, and I feel that the proof is correct given your additional explanations. I really appreciate the theoretical contribution of the paper. While I have changed my score, I hope that you can add all the missing arguments to your paper and spend time improving the presentation. The results are important but the current presentation may prevent many readers from appreciating the true value of the results. I have also spent some time discussing your paper with the area chair. The area chair may provide some additional suggestions to your paper, which would be very helpful. Please do pay attention to the meta review and follow the area chair's suggestions.

---

> > > ### Author Response · Authors · 2021-08-24
> > > **Including clarification argument**
> > >
> > > As we promised we will add the detailed argument to the paper. We are looking forward to the meta review and additional suggestions about how to make the presentation more clear. Thank you again for the detailed review and insightful comments!

---

### Decision · Program_Chairs · 2021-09-27

**Decision:**

Accept (Poster)

**Comment:**

The paper studies model selection in contextual bandits (and related problems) and the main result is a lower bound showing that "proper" algorithms cannot obtain the model selection guarantee conjecture in [FKL20]. A consequence is that the second order bound conjecture by Freund in a COLT open problem is not achievable.

The result is very interesting and after a careful reading along with discussions with the reviewers, we believe it is correct. It is clearly a very important result and represents significant progress in our understanding of online learning and bandit problems. As such the reviewers and I agree the paper should be accepted.

However, the paper was very hard to read, particularly regarding the proof of the lower bounds. My thoughts/suggestions here:
- I do not think the calculations for wrapping up the proof are very illuminating, e.g., paragraph block labeled "Proof of theorem 3" and "proof of theorem 4." I would prefer to see these replaced with a more qualitative/conceptual explanation of what is going on in the proof. Obviously the calculations should be included in the appendix, but I feel something more conceptual might be helpful to convince a reader.
- I found it helpful to think about what an algorithm might do to convince myself. For example, for Theorem 3 the algorithm basically has two options: (1) it can choose to play only arm i=K or (2) it can essentially explore uniformly in arms [K-1] for N_max rounds. In the former case, the environment will not switch, which is a contradiction of the algorithm's guarantee, while in (2) it incurs a lot of regret due to Lemma 1. Then you can do the calculation showing how you set \Delta to ensure that both of these cases work out to "confirm" the theorem. This is not a proof but I think provides more intuition than the calculations provided at present.

For the stochastic lower bound I think it is worth highlighting the following steps (this is also how I understand the proof):
- A proper learner means that there is no "cherry-picking" on contexts, so the statistical lower bound is a "passive learning" lower bound
- The construction is set up so that the lower bound part is essentially full information. In particular the algorithm always knows the loss for action 3 and the realized losses for actions 1 and 2 are coupled so that when you "explore" you are in a full information setting.
- Thus we can reduce to "standard" full-information passive learning problems. The flavor of such problem is closely related to work on "estimating learnability" [1] or "tolerant testing" [2].  This is a very clever reduction and highlights the role of the proper learning assumption.
- The specific learning problem being reduced to is essentially the problem of "estimating sparsity" in linear regression (see also the part on learning dictators in [2]), in particular where the sparsity is just 1. In fact I think Lemma 2 is very similar to Theorem 4.5 in [3] (although that theorem/paper is also quite hard to read). It would be great to mention this, so that readers who know this work (or who trust that result) can quickly convince themselves that your result is correct.
- Actually the results in [3] demonstrate something quite interesting that I think would be worth pointing out here. If I understand it correctly, their results show that a slightly different construction does not completely work for the model selection lower bound. Suppose that the means are given as in your construction, but rather than Bernoulli noise, the noise is Gaussian. Then the statistic \frac{1}{n} \sum_i (\ell_{i,1} - 1/2)^2 actually provides a better testing sample complexity bound than the lower bound in Lemma 2. The key here is that the variances under both hypothesis are the same, so the difference in mean can be detected by looking at the second moment. On the other hand, the Bernoulli construction is closer to the "unknown variance" setting in [3] where the mean difference is washed out by changing the variance. (You can see that in [3] the complexity of the "known variance" and "unknown variance" problems are actually different.)
- This is more or less how I convinced myself that the proof is correct.

At a high level, I think it would be great to re-write the lower bound section to capture the following two properties: (1) some hand-wavy way to convince an expert that the result is likely correct, without them having to read the technical details (e.g., by citing related results and maybe following the recipe above), and (2) some intuition based more on algorithmic considerations which I think are easier for readers to understand conceptually.

Finally, regarding proper vs improper algorithms. The reviewer raised a concern here on the actual definition but additionally there are many contextual bandit algorithms that are formally improper (e.g., mixing uniform exploration makes an algorithm improper). Additionally, maybe you can use the lower bound argument on active tolerant testing of dictators in [2] to actually get a lower bound against improper algorithms? Otherwise I think it is fine to conjecture this, but I might slightly tone down the claims about solving the open problems in [FKL20]. If I understand it correctly, the present paper formally resolves Open Problem 1 (and 1a, 1b) in [FKL20] but does not formally resolve open problem 2 due to the proper/improper learning issue. I am aware of some some folklore constructions that critically rely on cherry-picking/active learning/improperness to obtain model selection guarantees in some special cases, so I personally would be more hesitant to make these claims.

In addition to my suggestions, I hope the authors incorporate feedback from the reviewers, who I know spent quite a lot of time studying the paper and understanding the details.

References:
1. Kong, Valiant. Estimating learnability in the sublinear data regime. https://arxiv.org/abs/1805.01626
2. Balcan, Blais, Blum, Yang. Active tolerant testing. http://www.cs.cmu.edu/~ninamf/papers/active-testing.pdf
3. Ingster, Tsybakov, Verzelen. Detection boundary in sparse regression. https://arxiv.org/abs/1009.1706
Please also follow references to find the related papers to these. I know there are others.